# Minimal Impact ControlNet: Advancing Multi-ControlNet Integration

**Shikun Sun**[1][*] **Min Zhou**[2]**, Zixuan Wang**[1]**, Xubin Li**[2]**, Tiezheng Ge**[2]**, Zijie Ye**[1]**,
Xiaoyu Qin**[1][†]**, Junliang Xing**[1]**, Bo Zheng**[2]**, Jia Jia**[1,3,4][†]

[1]Department of Computer Science and Technology, Tsinghua University
[2] Taobao & Tmall Group of Alibaba
[3]BNRist, Tsinghua University
[4]Key Laboratory of Pervasive Computing, Ministry of Education
`{ssk21,wangzixu21}@mails.tsinghua.edu.cn, yzjscwy@gmail.com`
`{yunqi.zm,lxb204722,tiezheng.gtz,bozheng}@alibaba-inc.com`
`{xyqin,jlxing,jjia}@tsinghua.edu.cn`

## Abstract

With the advancement of diffusion models, there is a growing demand for high-quality, controllable image generation, particularly through methods that utilize one or multiple control signals based on ControlNet. However, in current Control-Net training, each control is designed to influence all areas of an image, which can lead to conflicts when different control signals are expected to manage different parts of the image in practical applications. This issue is especially pronounced with edge-type control conditions, where regions lacking boundary information often represent low-frequency signals, referred to as silent control signals. When combining multiple ControlNets, these silent control signals can suppress the generation of textures in related areas, resulting in suboptimal outcomes. To address this problem, we propose Minimal Impact ControlNet. Our approach mitigates conflicts through three key strategies: constructing a balanced dataset, combining and injecting feature signals in a balanced manner, and addressing the asymmetry in the score function's Jacobian matrix induced by ControlNet. These improvements enhance the compatibility of control signals, allowing for freer and more harmonious generation in areas with silent control signals.

## 1 Introduction

Recent advancements in diffusion models (Sohl-Dickstein et al., 2015; Ho et al., 2020; Rombach et al., 2022; Podell et al., 2023) have significantly bolstered the field of image generation. These innovations are particularly notable for the incorporation of controlled generation techniques, such as ControlNet (Zhang et al., 2023) and IP-Adapter (Ye et al., 2023), which allow precise manipulations using one or more control signals. Despite these advancements, challenges remain, particularly when integrating multiple control signals.

The primary difficulty arises from the fact that during the training of ControlNet, each control is designed to influence all areas of an image. This can lead to conflicts when different control signals are expected to manage different parts of the image in practical applications. This issue is especially pronounced with edge-type control conditions, where regions lacking boundary information often represent low-frequency signals, referred to as **silent control signals** by us.

As shown in Figure 1, our observations suggest that when combining multiple ControlNets, these silent control signals can suppress the generation of textures in areas where other control signals aim to generate details, resulting in suboptimal outcomes. This not only compromises the fidelity of the generated images but also restricts the flexibility and effectiveness of the control mechanisms within the model.

---

[*]Work done when Shikun Sun was an intern at Taobao & Tmall Group of Alibaba.
[†]Corresponding author

To tackle these challenges, adhering to the principle of "less is more", we introduce the Minimal Impact ControlNet (**MIControlNet**), a novel framework designed to refine the integration of multiple control signals within diffusion models. Our approach includes strategic modifications to the training data to reduce biases and utilizes a multi-objective optimization strategy during the feature combination phase, as well as addressing the asymmetry in the score function's Jacobian matrix induced by ControlNet. These methods aim to minimize conflicts between different control signals and between control signals and the inherent features of the dataset, thereby ensuring better compatibility and fidelity in the generated images.

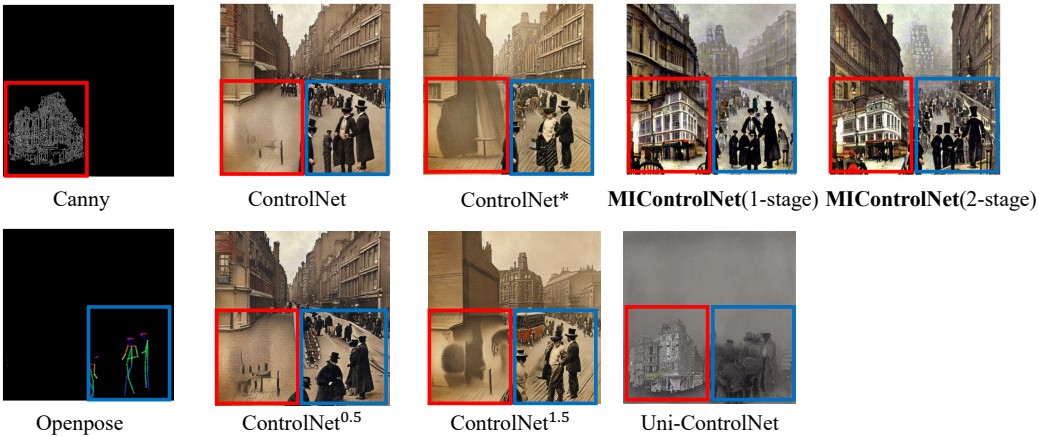

Figure 1: The **silent control signal** from OpenPose ControlNet (outside the blue box) suppresses the high-frequency control signal from Canny ControlNet (inside the red box). The black regions of the control signals represent the silent control signals.

To summarize, our main contributions are three-fold as follows:

- **Introduce silent control signals:** First introduce silent control signals that should remain inactive when other control signals are engaged, improving the compactness of the generation.

- **Feature injection and combination:** Employ strategies based on multi-objective optimization principles to improve model performance.

- **Theoretical contribution:** Develop and integrate a conservativity loss function within a large modular network architecture to ensure more stable learning dynamics.

By addressing the fundamental issues in training data preparation and signal integration, MIControlNet enhances the model's ability to follow the correct control signals in areas previously affected by control signal conflicts. Additionally, it improves controllability in high-frequency regions.

## 2 PRELIMINARIES

### 2.1 DIFFUSION MODELS FOR TEXT TO IMAGE GENERATION

Diffusion Models (Sohl-Dickstein et al., 2015; Ho et al., 2020; Rombach et al., 2022; Podell et al., 2023; Song et al., 2021) has gain great success as a generative models, especially in the text to image generation task (Rombach et al., 2022; Podell et al., 2023).

Suppose the image data distribution is $q(\mathbf{x}) = q_0(\mathbf{x}_0)$, where $\mathbf{x} \in \mathcal{X} \subset \mathcal{R}^{CHW}$. We define a forward process through a sequence of distributions, $q_t(\mathbf{x}_t) = \mathcal{N}(\alpha_t \mathbf{x}_0, (1 - \alpha_t^2)\mathbf{I})$, with $\{\alpha_t\}$ decreasing for $t \in [0, T] \cap \mathbb{Z}$. Here, $\alpha_0 = 1$ and $\alpha_T \approx 0$. In the generation process, we initiate from $\mathbf{x}_T \sim \mathcal{N}(\mathbf{0}, \mathbf{I})$ and iteratively generate a sample of the previous timestep using a denoising network $\epsilon_\phi(\mathbf{x}_t, t)$, trained by minimizing the prediction of added noise as follows:

$$\mathbb{E}_{\mathbf{x}_0 \sim q_0(\mathbf{x}_0), t, \epsilon \sim \mathcal{N}(\mathbf{0}, \mathbf{I})} w(t) \|\epsilon_\phi(\alpha_t \mathbf{x}_0 + \sigma_t \epsilon, t) - \epsilon\|_2^2, \tag{1}$$

where $w(t)$ balances the losses across different timesteps, $t$ is uniformly selected from $0$ to $T$, and $\sigma_t = \sqrt{1 - \alpha_t^2}$. This loss also serves as the learning objective for numerous control methods, such as ControlNet as described by Zhang et al. (2023).

Researchers Song et al. (2021); Vincent (2011) have developed the theory of score-based diffusion models. They established a connection between the score of $q_t$ and $\epsilon_\phi(\mathbf{x}_t, t)$ as:

$$\mathbf{s}\left(\mathbf{x}_t, t\right) = \nabla_{\mathbf{x}_t} \log q_t(\mathbf{x}_t, t) \approx -\frac{\epsilon_\phi(\mathbf{x}_t, t)}{\sigma_t}. \tag{2}$$

## 2.2 CONTROLNET

ControlNet (Zhang et al., 2023) marks a substantial breakthrough in controlled generation for diffusion models, utilizing low-level features such as edges, poses, and depth maps to refine the generative process. The original ControlNet Zhang et al. (2023) paper and Figure 2 provide a more intuitive explanation through graph; here, we opt for a formulaic approach to introduce symbols that facilitate the proofs in subsequent sections.

Consider a U-Net architecture where the encoder $\mathcal{E}^\theta$ and decoder $\mathcal{D}^\psi$ consist of layers $\{\mathcal{E}_i^\theta\}_{i=1}^{l+1}$ and $\{\mathcal{D}_i^\psi\}_{i=l}^1$, respectively, with $i$ indicating the layer index and $\theta, \psi$ representing the model parameters. The architecture of ControlNet, $\mathcal{C}^\phi = \{\mathcal{C}_i^\phi\}_{i=1}^{l+1}$, parallels that of $\mathcal{E}^\theta$ but also integrates a control image as an input.

Ignoring the input and output layers, we start with $\mathbf{f}_0^e = \mathbf{x}$ and $\mathbf{f}_0^c = \mathbf{x} + \mathbf{conv}(\mathbf{c})$. The input of $\mathcal{E}_i$, $\mathcal{D}_i$ and $\mathcal{C}_i$ are $\mathbf{f}_i^e, \mathbf{f}_c^e$ and $\{\mathbf{f}_{i+1}^{dres}, \mathbf{add}\left(\mathbf{f}_i^{eres}, \mathbf{f}_i^{cres}\right)\}$; the output of $\mathcal{E}_i$, $\mathcal{D}_i$ and $\mathcal{C}_i$ are $\{\mathbf{f}_{i+1}^e, \mathbf{f}_{i+1}^{eres}\}$, $\mathbf{f}_i^d$ and $\{\mathbf{f}_{i+1}^c, \mathbf{f}_{i+1}^{cres}\}$ respectively, where $\mathbf{f}^e, \mathbf{f}^d, \mathbf{f}^c$ are the direct outputs and $\mathbf{f}^{eres}$ and $\mathbf{f}^{cres}$ are the residual outputs.

For the convenience of calculating the Jacobian matrix of the score function, we clearly define the components of the whole encoder $\mathcal{E}$, ControlNet $\mathcal{C}$, and decoder $\mathcal{D}$ as follows:

$$\mathcal{E}(\mathbf{x}) = (\mathbf{f}_1^{eres}, \mathbf{f}_2^{eres}, \dots, \mathbf{f}_{l+1}^{eres}), \tag{3}$$

$$\mathcal{C}(\mathbf{x}, \mathbf{c}) = (\mathbf{f}_1^{cres}, \mathbf{f}_2^{cres}, \dots, \mathbf{f}_{l+1}^{cres}), \tag{4}$$

and

$$\mathcal{D}(\mathbf{f}_1^d, \mathbf{f}_2^d, \dots, \mathbf{f}_{l+1}^d) = \mathbf{s}, \tag{5}$$

where $\mathbf{f}_i^d = \mathbf{add}\left(\mathbf{f}_i^{eres}, \mathbf{f}_i^{cres}\right)$ and $\mathbf{s}$ represents the score function.

## 2.3 SCORE FUNCTION AND ITS CONSERVATIVITY

During the parameterization of the score function $\mathbf{s}(\mathbf{x}_t, t)$ with a neural network, there are no inherent constraints on the conservativity of score functions (Salimans & Ho, 2021). The conservativity of a vector field indicates that the field can be represented as the gradient of a scalar-valued function. For instance, the score function $\mathbf{s}(\mathbf{x}_t, t)$ can be modeled as the gradient of $\log q_t(\mathbf{x}_t, t)$. A straightforward method to verify if a vector field is conservative involves computing the Jacobian matrix of the vector field and checking if this matrix is symmetric. This symmetry is a consequence of the commutativity of the partial derivatives of the scalar function.

However, directly calculating the Jacobian matrix of the score function poses significant challenges. As an alternative, we utilize stochastic estimators and leverage the capabilities of modern neural network frameworks, such as the vector-Jacobian computation features in PyTorch, to obtain an unbiased estimation of the trace of the Jacobian matrix and related values.

To enforce the conservativity of the score function directly, suppose the Jacobian matrix of $\mathbf{s}_t$ with respect to $\mathbf{x}_t$ is denoted as $\mathbf{J}_{\mathbf{s}_t, \mathbf{x}_t}$. We propose using the following loss function:

$$\mathcal{L}_{QC} = \frac{1}{2} \mathbb{E}_{t, \mathbf{x}_t} \left\| \mathbf{J}_{\mathbf{s}_t, \mathbf{x}_t} - \mathbf{J}_{\mathbf{s}_t, \mathbf{x}_t}^\mathsf{T} \right\|_F^2, \tag{6}$$

where $F$ represents the Frobenius norm. That formula can be equivalently expressed as (Chao et al., 2022):

$$\mathcal{L}_{QC} = \mathbb{E}_{t,\mathbf{x}_t} \left[ \text{tr}(\mathbf{J}_{\mathbf{s}_t,\mathbf{x}_t} \mathbf{J}_{\mathbf{s}_t,\mathbf{x}_t}^{\mathsf{T}}) - \text{tr}(\mathbf{J}_{\mathbf{s}_t,\mathbf{x}_t} \mathbf{J}_{\mathbf{s}_t,\mathbf{x}_t}) \right], \tag{7}$$

where the trace of the product of Jacobian matrices can be efficiently estimated using Hutchinson's trace estimator (Hutchinson, 1989). However, even such an estimator can be computationally expensive, especially when dealing with large-scale neural networks.

## 2.4 MULTI-OBJECTIVE OPTIMIZATION

The goal of multi-objective optimization is to find the Pareto optimal solution, a state where no single objective can be improved without degrading others. Similar to single-objective optimization, local Pareto optimality can also be achieved using gradient descent techniques. The Multiple Gradient Descent Algorithm (MGDA) (Désidéri, 2012), is one such method for attaining local Pareto optimal solutions. The central concept of MGDA is to balance all gradients towards a direction that forms acute angles with each gradient, thereby ensuring that no objective worsens as a result of an optimization step. It also has applications in image and 3D generation tasks (Sun et al., 2023; Huang et al., 2024). We think the idea of forming an acute angle with each gradient is a good way to balance the gradients, and we will use this idea in the feature combination and feature injection.

## 3 PROBLEMS IN MULTI-CONTROLNET COMBINATION

Our problem setting aligns with the current standard in the community, wherein each ControlNet is trained individually. At the sampling stage, these networks are combined according to different control signals, functioning as plug-ins. This setup ensures flexibility and modularity, allowing for the seamless integration of various control signals to enhance the model's generative capabilities. However, this approach has several limitations, particularly when combining multiple ControlNets.

**Data Bias in Areas with "Silent" Control Signals.** In the training of ControlNet models, a significant issue arises from the presence of "silent" control signals, particularly with edge condition signals. These "silent" control signals are characterized by empty conditions where the corresponding paired image areas are often blurred or lack high-frequency information. This leads to a data bias during training, causing the model to suppress high-frequency information in the generated images. While this suppression can be advantageous for strict generations in single-control scenarios, it poses a challenge in multi-ControlNet combination scenarios. When two control signals coexist in an area—one with high-frequency information and the other being a "silent" control signal—a conflict arises. The model, influenced by the "silent" control signal, may undesirably suppress high-frequency information in the generated images. This conflict is problematic in multi-ControlNet combination scenarios, where the preservation of high-frequency details is crucial.

**Optimal Ratios for Multi-ControlNet Combination.** Another challenge in combining multiple ControlNets is determining the optimal ratios for merging various control signals. There is currently no clear guideline for the combination of different control signals, making this process difficult. The current practice involves combining control signals as plug-ins at the sampling stage, relying on user experimentation. This approach may lead to suboptimal results, as the model may not effectively balance the different control signals.

**Conservativity of Conditional Score Function.** Although the conservativity of diffusion models is well researched, the situation differs for ControlNet, where another network is tuned to control the diffusion model with much less data compared to the original diffusion model. Therefore, it is essential to consider the conservativity of the enhanced score function when combining multiple ControlNets to ensure stability or seek improved performance.

## 4 MINIMAL IMPACT CONTROLNET

To address the aforementioned issues, we introduce the MIControlNet. The key idea of MIControlNet is to minimize the impact of the ControlNet on the original U-Net by reducing the conflicts of

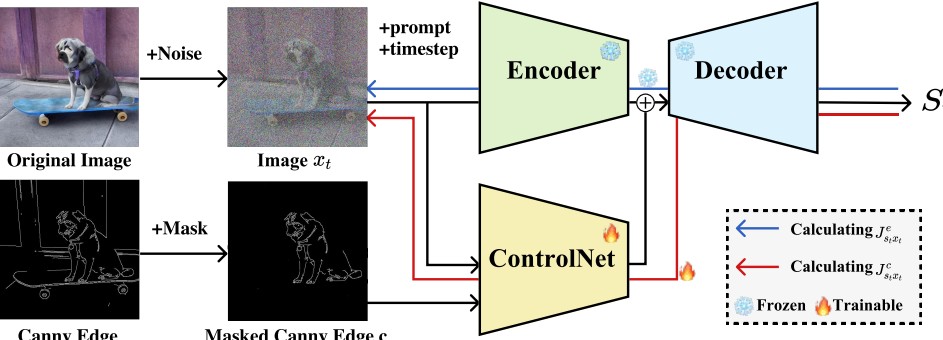

Figure 2: Overview of our data flow. Masking specific areas of the conditions allows the silent control signals to generate more diverse patterns, while exhibiting reduced controllability when interacting with high-frequency control signals.

each ControlNet, thereby achieving the best combination of multiple ControlNets. We will introduce the details of MIControlNet in the following sections following the order of problem setting, training strategies and sampling strategies.

## 4.1 REBALANCE THE DISTRIBUTION

To solve the first problem, we apply simple but effective data augmentation techniques to rebalance the distribution of areas lacking control signals. Specifically, we will apply segmentation masks from images on the control signals, enhancing the diversity of the image areas corresponding to the silent control signals, which is the same as inpainting the image area with the silent control signals. This process will help the model learn to generate high-frequency information in areas with silent control signals, thereby reducing data bias during training. Further details will be explained in the Appendix F.1.

## 4.2 MINIMAL IMPACT ON FEATURE INJECTION AND COMBINATION

To minimize the impact of the ControlNet signal of each layer $\mathbf{f}_i^{cres}$ on the original U-Net encoder feature $\mathbf{f}_i^{eres}$, we draw inspiration from the MGDA algorithm Désidéri (2012), which involves forming acute angles with each vector. We employ a restricted MGDA-based balancing algorithm to regulate the injection of control signals into each layer, which will keep the coefficient of original feature $\mathbf{f}_i^{eres}$. In detail, we will apply the following new dynamic $\mathbf{add}^{inj}$ function to the feature injection process.

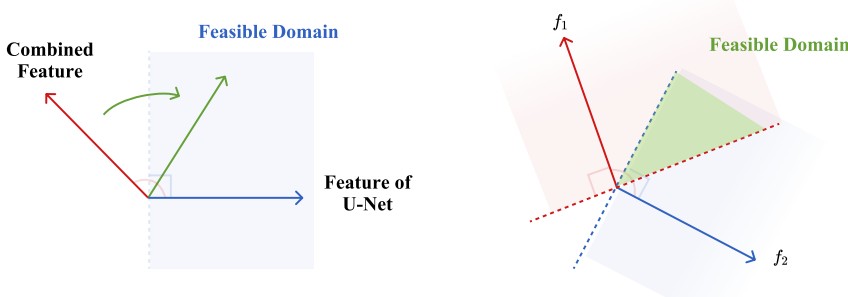

Figure 3: The left image shows the feature injection process in MIControlNet, while the right image illustrates the feature combination process. The Feasible Domain is where the combined optimization direction aligns with the U-Net feature or both control signal features $f_1$ and $f_2$.

Firstly, we calculate the coefficient of injected control signal $\lambda_i$ for each layer $i$ as follows:

$$\lambda_i^*(\mathbf{v}_1, \mathbf{v}_2) = \min \left[ 1, \max \left[ \frac{(\mathbf{v}_2 - \mathbf{v}_1)^T \mathbf{v}_2}{\|\mathbf{v}_2 - \mathbf{v}_1\|_2^2}, 0 \right] \right], \tag{8}$$

and then

$$\lambda_i(\mathbf{v}_1, \mathbf{v}_2) = \frac{\lambda_i^*(\mathbf{v}_1, \mathbf{v}_2)}{1 - \lambda_i^*(\mathbf{v}_1, \mathbf{v}_2)}. \tag{9}$$

The new **add** function is defined as follows:

$$\mathbf{add}^{inj}(\mathbf{f}_i^{eres}, \mathbf{f}_i^{cres}) = \mathbf{f}_i^{eres} + \lambda_i(\mathbf{f}_i^{eres}, \mathbf{f}_i^{cres}) \cdot \mathbf{f}_i^{cres}. \tag{10}$$

The key constraint we impose is to maintain the coefficient of $\mathbf{f}_i^{\text{eres}}$ at 1, ensuring the preservation of the original U-Net data flow architecture. Practically, the range of $\lambda_i$ is limited to [0, 20] to mitigate the risk of overpowering control signals that could suppress the original features. Further details on the connection to MGDA are elaborated in the Appendix E.

To balance different control signals during the sampling stage, we also utilize the concept of forming an acute angle with each gradient, which initially balances the various control signals before the injection.

For the feature maps associated with different control signals, denoted as $\mathbf{f}_i^{cres,1}$ and $\mathbf{f}_i^{cres,2}$, we employ a combination strategy defined by the following equation:

$$\mathbf{f}_i^{cres} = \mathbf{add}^{com}(\mathbf{f}_i^{cres,1}, \mathbf{f}_i^{cres,2}), \tag{11}$$

where the $\mathbf{add}^{com}$ function is explicitly defined as:

$$\mathbf{add}^{com}(\mathbf{f}_i^{cres,1}, \mathbf{f}_i^{cres,2}) = (1 - \lambda_i^*(\mathbf{f}_i^{cres,1}, \mathbf{f}_i^{cres,2}))\mathbf{f}_i^{cres,1} + \lambda_i^*(\mathbf{f}_i^{cres,1}, \mathbf{f}_i^{cres,2})\mathbf{f}_i^{cres,2}. \tag{12}$$

Then we follow the same process as in the feature injection stage to calculate the coefficient of the combined feature map $\lambda_i$ for each layer $i$.

### 4.3 MINIMAL IMPACT ON CONSERVATIVITY

To estimate the $\mathcal{L}_{QC}$ in Eqn. 7, we need to apply the Hutchinson's estimator to the Jacobian matrix of the model. However, such estimation still needs to construct second order derivatives, which is computationally expensive.

Our insight is that the parameters of ControlNet primarily manage the additional conservativity it introduces. By decomposing the Jacobian matrix of the model into two components, we can isolate and only calculate the conservativity loss specific to ControlNet, which follows the red line in Figure 2, making the process simpler and more efficient. Additionally, applying this conservativity loss ensures control over all the extra unconservativity introduced by ControlNet.

Suppose $\mathbf{v}$ fits a distribution whose expectation is $\mathbf{0}$ and variance is $\mathbf{I}$, by the Hutchinson's estimator, we have a unbiased estimation of $\mathcal{L}_{QC}$, which is

$$\mathcal{L}_{QC}^{est} = \mathbb{E}_{\mathbf{v},t,\mathbf{x_t}} \left[ \mathbf{v}^\mathsf{T} \mathbf{J}_{\mathbf{s}_t,\mathbf{x}_t} \mathbf{J}_{\mathbf{s}_t,\mathbf{x}_t}^\mathsf{T} \mathbf{v} - \mathbf{v}^\mathsf{T} \mathbf{J}_{\mathbf{s}_t,\mathbf{x}_t} \mathbf{J}_{\mathbf{s}_t,\mathbf{x}_t} \mathbf{v} \right]. \tag{13}$$

We propose the following proposition to decompose the Jacobian matrix of the model into the original Jacobian matrix from the U-Net and the additional Jacobian matrix introduced by ControlNet.

**Proposition 4.1 (Decomposition of Jacobian Matrix)** *In a U-Net model augmented with ControlNet, the overall Jacobian matrix $\mathbf{J}_{\mathbf{s}_t,\mathbf{x}_t}$ can be decomposed into the original Jacobian matrix $\mathbf{J}_{\mathbf{s}_t,\mathbf{x}_t}^e$ from the U-Net and an additional Jacobian matrix $\mathbf{J}_{\mathbf{s}_t,\mathbf{x}_t}^c$ introduced by ControlNet:*

$$\mathbf{J}_{\mathbf{s}_t,\mathbf{x}_t} = \mathbf{J}_{\mathbf{s}_t,\mathbf{x}_t}^e + \mathbf{J}_{\mathbf{s}_t,\mathbf{x}_t}^c. \tag{14}$$

And due to the large training data gap between traing the U-Net and ControlNet, we propose the following assumption to depart them in parameters level.

**Assumption 4.1 (Responsibility for Conservativity)** *In the U-Net model equipped with Control-Net, the conservativity of the original Jacobian matrix is governed by the parameters of the U-Net. Meanwhile, the parameters of ControlNet are principally tasked with managing the additional conservativity introduced by ControlNet, which can be described as*

$$\nabla_\phi \mathbf{J}^e_{\mathbf{s}_t, \mathbf{x}_t} = \mathbf{0}, \tag{15}$$

*where $\phi$ is the parameters of ControlNet.*

Then, we can ignore the parts does not containing $\mathbf{J}^c_{\mathbf{s_t}, \mathbf{x_t}}$ in Eqn. 13 and got a new loss for optimization of the ControlNet, which is

$$\mathcal{L}^c_{QC} = \mathbb{E}_{\mathbf{v}, t, \mathbf{x_t}} \mathbf{v}^\top \left[ 2\mathbf{J}^e_{\mathbf{s}_t, \mathbf{x}_t} \mathbf{J}^{c\top}_{\mathbf{s}_t, \mathbf{x}_t} - 2\mathbf{J}^e_{\mathbf{s}_t, \mathbf{x}_t} \mathbf{J}^c_{\mathbf{s}_t, \mathbf{x}_t} + \mathbf{J}^c_{\mathbf{s}_t, \mathbf{x}_t} \mathbf{J}^{c\top}_{\mathbf{s}_t, \mathbf{x}_t} - \mathbf{J}^c_{\mathbf{s}_t, \mathbf{x}_t} \mathbf{J}^c_{\mathbf{s}_t, \mathbf{x}_t} \right] \mathbf{v}. \tag{16}$$

We have

**Proposition 4.2** *Under Assumption 4.1, the gradient of the conservativity loss of ControlNet is equal to the gradient of the estimated conservativity loss, which is given by*

$$\nabla_\phi \mathcal{L}^c_{QC} = \nabla_\phi \mathcal{L}^{est}_{QC}. \tag{17}$$

However, due to computation limitations, we still want to fully remove the $\mathbf{J}^e_{\mathbf{s}_t, \mathbf{x}_t}$ term. We have the following simplified loss for the ControlNet optimization, which is

$$\mathcal{L}^{simple}_{QC} = \mathbb{E}_{\mathbf{v}, t, \mathbf{x_t}} \mathbf{v}^\top \left[ \mathbf{J}^c_{\mathbf{s}_t, \mathbf{x}_t} \mathbf{J}^{c\top}_{\mathbf{s}_t, \mathbf{x}_t} - \mathbf{J}^c_{\mathbf{s}_t, \mathbf{x}_t} \mathbf{J}^c_{\mathbf{s}_t, \mathbf{x}_t} \right] \mathbf{v}. \tag{18}$$

And we have the following proposition for the relationship between the simplified loss and the original loss.

**Theorem 4.1** *Suppose the Frobenius norm of $\mathbf{J}^e_{\mathbf{s}_t, \mathbf{x}_t}$ is uniformly bounded by $M$, we have*

$$\mathcal{L}^c_{QC} \le 2\sqrt{2}M\sqrt{\mathcal{L}^{simple}_{QC}} + \mathcal{L}^{simple}_{QC}, \tag{19}$$

*which indicates that if the simplied loss is zero, the original loss is also zero.*

In practice, we just apply the simplified loss to optimize the ControlNet.

## 5 EXPERIMENTS

### 5.1 EXPERIMENT SETUP

**Dataset.** For training, we primarily use the MultiGen-20M dataset (Qin et al., 2023), a subset of LAION-Aesthetics (Schuhmann et al., 2022), which provides conditions such as Canny (Canny, 1986), Hed (Xie & Tu, 2015), and OpenPose (Cao et al., 2017). This dataset also includes segmentations, which facilitate balancing the ground truth of areas with silent control signals. For evaluation, we randomly sample images from LAION-Aesthetics and use them and their extracted conditions. For single control signal, we use the original prompts from the dataset. For multi control signals, we directly concate the corresponding prompts as the prompts.

**Implementation.** We initially train our model using balanced data and feature injection for the Canny, HED, Depth, and OpenPose conditions. Once the model converges, we label this phase as 1-stage. We then continue training the model with an additional conservativity loss, labeling this phase as 2-stage. Further details are provided in the Appendix. F. For sampling, we apply balanced feature injection and combination in our models. More results are in the Appendix H.

**Baseline.** We select the newest ControlNet v1.1 and Uni-ControlNet (Zhao et al., 2024) as our baseline. For the ControlNet baseline, we provide both the original feature combination and a balanced version, labeled as ControlNet*. We also introduce fixed scaling factors for the first control signals, labeled ControlNet$^{0.5}$ and ControlNet$^{1.5}$, maintaining a total scaling factor of 2.0 to match the original ControlNet and our feature combination method. Additionally, we include Control-Net**, which is trained with the same data augmentation as our method.

Some other models, such as ControlNet++ (Li et al., 2024), are optimized for precise control, making them less suitable as baselines in our experimental settings.

## 5.2 SINGLE CONTROL SIGNAL

In this subsection, we primarily examine the improvements our method brings when using a single control signal. The main improvement lies in the inpainting ability of the silent control signal.

### 5.2.1 QUALITATIVE COMPARISON

We mainly conduct two qualitative comparisons:

**Total Variance under Silent Control Signals.** For the calculation of the total variance, we sample 500 images from the LAION-Aesthetics dataset and calculate the total variance in the regions controlled by the silent control signals. We then compare the results with ControlNet. As shown in Figure 4a, the results indicate the ability of our method to generate more diverse texture patterns in these situations.

**Asymmetry in the Jacobian Matrix.** We analyze the asymmetry of the extra part of the Jacobian matrix introduced by ControlNet, as discussed in *Asym* metric defined by Chao et al. (2022). This indicates the asymmetry of the Jacobian matrix. As shown in Figure 4b, our method reduces the asymmetry, leading to more stable and consistent control. The *Asym* metric is estimated on the MultiGen-20M dataset, using a batch size of 64 for 100 steps. We observed that after the second-stage training, the asymmetry introduced by ControlNet significantly diminishes, indicating a smaller impact on the original U-Net, There is also an interesting phenomenon where the decreases in *Asym* are similar on a logarithmic scale.

We also compare the FID and convergence speed as described in Appendix G. Our model achieves similar image quality with faster convergence compared to the ControlNet baseline.

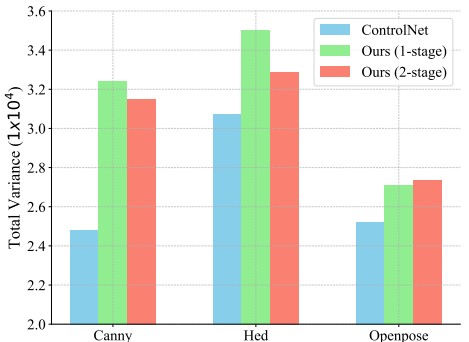

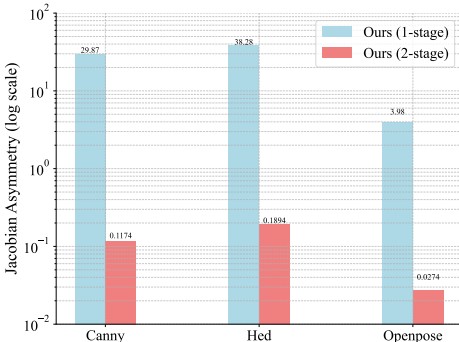

(a) Total variance under silent control signals.  (b) Asymmetry in the Jacobian Matrix.

Figure 4: Two qualitative comparisons for single control signal.

### 5.2.2 VISUAL COMPARISON

We compare the visual results for single control signals of the ControlNet and our MIControlNet in Figure 5. More results are in the Appendix H. Our method demonstrates the ability to generate more texture patterns in areas corresponding to silent control signals, which aligns with the quantitative results shown in Figure 4a.

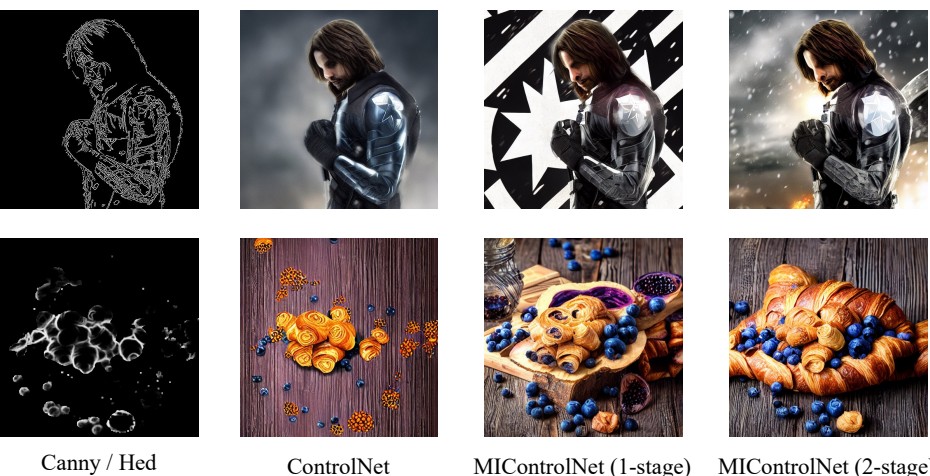

| Canny / Hed | ControlNet | MIControlNet (1-stage) | MIControlNet (2-stage) |

Figure 5: Comparision of ControlNet and MIControlNet for single condition generation.

## 5.3 MULTI-CONTROL SIGNALS

In this subsection, we examine the improvements introduced by our method when using multiple control signals. We randomly selected 2,000 images from the LAION-Aesthetics dataset and extracted the central portion of two conditions in equal measure for sampling. These conditions were randomly resized and placed on either the left or right side, with the remaining area filled by silent signals. We then used these modified control signals to generate images. To save space, we present both conditions in a single image in Figure 7.

Table 1: The FID of the multi-condition scenario. Each condition is associated with its own FID. the FID scores are presented with the best result highlighted in bold and the second best underlined.

| Methods | Openpose-Canny | Openpose-Hed | Canny-Hed | Hed-Depth |
|---------|----------------|--------------|-----------|-----------|
| ControlNet | 80.37 / 111.30 | 76.98 / 84.20 | 123.59 / 86.43 | 91.98 / 86.25 |
| ControlNet$^{0.5}$ | 105.86 / 123.13 | 145.88 / 107.52 | 143.67 / 106.40 | -/- |
| ControlNet$^{1.5}$ | **74.37** / 99.44 | 74.52 / 86.57 | 120.84 / 88.38 | -/- |
| ControlNet* | 77.43 / 89.57 | 76.69 / 78.31 | 122.10 / 85.45 | 78.14 / 90.65 |
| ControlNet** | 92.98 / 84.02 | 87.33 / 78.49 | 77.02 / 75.46 | 74.28 / 81.16 |
| Uni-ControlNet | 96.50 / 74.55 | 139.87 / 76.06 | 88.77 / 75.47 | 73.68 / 89.94 |
| Ours (1-stage) | 76.13 / 77.22 | **70.32** / **68.42** | 74.19 / 70.26 | 71.16 / 71.93 |
| Ours (2-stage) | 75.77 / **72.25** | 73.45 / 71.74 | **71.34** / **69.35** | **69.68** / **71.18** |

### 5.3.1 QUALITATIVE COMPARISON

**FIDs for each control signals.** We calculate the FIDs for two conditions in a multi-condition scenario. For each condition, we extract the relevant part of the generated image and compute the FID against the original 1,000 images. As shown in Table 1, our 2-stage MIControlNet achieves the best FIDs in most cases, indicating that our method outperforms the baselines and highlights its effectiveness in multi-condition scenarios. Our feature injection and combination technique achieves an average improvement of 9.79 over the vanilla ControlNet with silent control signal targeted data augmentation. The data augmentation alone achieves an average improvement of 11.26.

**Cycle consistency for each control signal.** Table 2 in Appendix G.2 shows the $L1$ distance between the extracted condition from the generated images and the original condition. Our MIControlNet achieves the lowest values in most cases, indicating better preservation of control signals (excluding silent control signals) in the generated images.

### 5.3.2 VISUAL COMPARISON

Visual comparisons are shown in Figure 7. In the first case, ControlNet fails to apply the openpose condition, while ControlNet* succeeds. In the second case, ControlNet fails to meet the canny condition. In the final case, ControlNet fails both control signals. Our method effectively silences silent control signals when other control signals are active, allowing the useful control signals to dominate. Additional visual results are provided in Appendix H.

## 6 CONCLUSION

In this paper, we introduced MIControlNet, designed to minimize the impact of ControlNet for improved multi-control signal integration. Our approach involves rebalancing the data distribution in areas controlled by silent control signals, introducing a multi-objective perspective to feature combination, and reducing the asymmetry in the Jacobian matrix of the score function. These strategies enhance the balance and compatibility of multiple ControlNets without necessitating joint training, enabling more free and harmonious generation using multiple control signals.

## 7 ACKNOWLEDGEMENTS

This work is funded by the National Key R&D Program of China under Grant No. 2024QY1400, the National Natural Science Foundation of China No. 62425604. This work is supported by Tsinghua University Initiative Scientific Research Program. This work is also supported by Alibaba Group through Alibaba Innovative Research Program. We sincerely thank Qixin Wang for her assistance in creating the pretty figures for this paper.

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

APPENDIX

## A  RELATED WORK

### A.1  IMAGE-BASED CONTROL METHODS FOR DIFFUSION MODELS

Image-based control methods are crucial for image generation. Following the success of diffusion models, numerous algorithms for controlled image generation have been developed, leading to the creation of techniques such as SDEdit (Meng et al., 2021), ControlNet (Zhang et al., 2023), and DreamBooth (Ruiz et al., 2023).

### A.2  CONSERVATIVITY IN DIFFUSION MODELS

With the significant success of score matching training algorithms in the unconstrained score approach, this method has become a focal point in research. The score functions learned in this manner are no longer conservative, meaning they may not strictly adhere to the constraints of the original data distribution. This lack of conservativity could impact model performance, and numerous studies have explored this phenomenon (Salimans & Ho, 2021; Chao et al., 2022; Horvat & Pfister, 2024; Lai et al., 2023). Researchers have attempted to adjust for this by incorporating either soft or hard conservativity constraints, producing some interesting theoretical results in the process.

However, while conservativity has been extensively studied in foundational models, there is a relative lack of research on conservativity in diffusion models that enhance control over generative capabilities through the addition of modules. Given the unique generation process of diffusion models, implementing effective conservativity controls is particularly critical, potentially offering new perspectives on improving model stability and generation quality.

## B  DISCUSSION AND LIMITATIONS

Compared with mainstream methods developed from ControlNet, which exert control influence across the entire image, our approach have distinct use cases. While mainstream ControlNet methods offer broad control capabilities, MIControlNet focuses on precise control in targeted areas, addressing conflicts arising from multiple control signals.

Our primary focus is on improving controllability. However, our method has not yet fully explored the potential of prompt engineering and related techniques, such as using negative prompts and sampling algorithms. There is significant room for improvement in these areas, which could further enhance the effectiveness and flexibility of controlled image generation.

The necessity of incorporating a conservativity loss is another crucial aspect of our approach. Due to resource constraints, we could not fully implement the conservativity loss in large-scale models. We hope future work will address this limitation, potentially leading to more robust implementations. Additionally, with the theoretical advancements in conservativity constraints, similar to the development of score matching, we anticipate the emergence of unbiased estimation algorithms for the trace Jacobian matrix that does not require second-order gradient backpropagation.

## C  BROADER IMPACT AND SAFEGUARDS

Generative AI has the potential to produce harmful information. To mitigate these risks, it is crucial to implement comprehensive safeguards. Accordingly, we will integrate a safety checker into our released code.

## D  PROOFS

### D.1  PROOF OF PROPOSITION 4.1

**Proposition D.1 (Decomposition of Jacobian Matrix)** *In a U-Net model augmented with Control-Net, the overall Jacobian matrix $\mathbf{J}_{\mathbf{s}_t, \mathbf{x}_t}$ can be decomposed into the original Jacobian matrix $\mathbf{J}^e_{\mathbf{s}_t, \mathbf{x}_t}$*

*from the U-Net and an additional Jacobian matrix $\mathbf{J}^c_{\mathbf{s}_t, \mathbf{x}_t}$ introduced by ControlNet:*

$$\mathbf{J}_{\mathbf{s}_t, \mathbf{x}_t} = \mathbf{J}^e_{\mathbf{s}_t, \mathbf{x}_t} + \mathbf{J}^c_{\mathbf{s}_t, \mathbf{x}_t}. \tag{20}$$

$$
\begin{aligned}
\mathbf{J}_{\mathbf{s}_t, \mathbf{x}_t} &= \sum_{i=1}^{l+1} \mathbf{J}_{\mathbf{s}_t, \mathbf{f}^d_i} \mathbf{J}_{\mathbf{f}^d_i, \mathbf{x}_t} \\
&= \sum_{i=1}^{l+1} \mathbf{J}_{\mathbf{s}_t, \mathbf{f}^d_i} \left[ \mathbf{J}_{\mathbf{f}^d_i, \mathbf{f}^{eres}_i} \mathbf{J}_{\mathbf{f}^{eres}_i, \mathbf{x}_t} + \mathbf{J}_{\mathbf{f}^d_i, \mathbf{f}^{cres}_i} \mathbf{J}_{\mathbf{f}^{cres}_i, \mathbf{x}_t} \right] \\
&= \sum_{i=1}^{l+1} \mathbf{J}_{\mathbf{s}_t, \mathbf{f}^d_i} \mathbf{J}_{\mathbf{f}^d_i, \mathbf{f}^{eres}_i} \mathbf{J}_{\mathbf{f}^{eres}_i, \mathbf{x}_t} + \mathbf{J}_{\mathbf{s}_t, \mathbf{f}^d_i} \mathbf{J}_{\mathbf{f}^d_i, \mathbf{f}^{cres}_i} \mathbf{J}_{\mathbf{f}^{cres}_i, \mathbf{x}_t} \\
&= \sum_{i=1}^{l+1} \mathbf{J}_{\mathbf{s}_t, \mathbf{f}^{eres}_i} \mathbf{J}_{\mathbf{f}^{eres}_i, \mathbf{x}_t} + \mathbf{J}_{\mathbf{s}_t, \mathbf{f}^{cres}_i} \mathbf{J}_{\mathbf{f}^{cres}_i, \mathbf{x}_t} \\
&= \sum_{i=1}^{l+1} \mathbf{J}_{\mathbf{s}_t, \mathbf{f}^{eres}_i} \mathbf{J}_{\mathbf{f}^{eres}_i, \mathbf{x}_t} + \sum_{i=1}^{l+1} \mathbf{J}_{\mathbf{s}_t, \mathbf{f}^{cres}_i} \mathbf{J}_{\mathbf{f}^{cres}_i, \mathbf{x}_t} \\
&= \mathbf{J}^e_{\mathbf{s}_t, \mathbf{x}_t} + \mathbf{J}^c_{\mathbf{s}_t, \mathbf{x}_t}.
\end{aligned}
\tag{21}
$$

## D.2 Proof of Proposition 4.2

**Proposition D.2** *Under Assumption 4.1, the gradient of the conservativity loss of ControlNet is equal to the gradient of the estimated conservativity loss, which is given by*

$$\nabla_\phi \mathcal{L}^c_{QC} = \nabla_\phi \mathcal{L}^{est}_{QC}. \tag{22}$$

$$
\begin{aligned}
\mathcal{L}^{est}_{QC} &= \mathbb{E}_{\mathbf{v}, t, \mathbf{x}_t} \left[ \mathbf{v}^{\mathsf{T}} \mathbf{J}_{\mathbf{s}_t, \mathbf{x}_t} \mathbf{J}^{\mathsf{T}}_{\mathbf{s}_t, \mathbf{x}_t} \mathbf{v} - \mathbf{v}^{\mathsf{T}} \mathbf{J}_{\mathbf{s}_t, \mathbf{x}_t} \mathbf{J}_{\mathbf{s}_t, \mathbf{x}_t} \mathbf{v} \right] \\
&= \mathbb{E}_{\mathbf{v}, t, \mathbf{x}_t} \mathbf{v}^{\mathsf{T}} \left[ 2\mathbf{J}^e_{\mathbf{s}_t, \mathbf{x}_t} \mathbf{J}^{c\mathsf{T}}_{\mathbf{s}_t, \mathbf{x}_t} - 2\mathbf{J}^e_{\mathbf{s}_t, \mathbf{x}_t} \mathbf{J}^c_{\mathbf{s}_t, \mathbf{x}_t} + \mathbf{J}^c_{\mathbf{s}_t, \mathbf{x}_t} \mathbf{J}^{c\mathsf{T}}_{\mathbf{s}_t, \mathbf{x}_t} - \mathbf{J}^c_{\mathbf{s}_t, \mathbf{x}_t} \mathbf{J}^c_{\mathbf{s}_t, \mathbf{x}_t} \right] \mathbf{v} \\
&\quad + \mathbb{E}_{\mathbf{v}, t, \mathbf{x}_t} \mathbf{v}^{\mathsf{T}} \left[ \mathbf{J}^e_{\mathbf{s}_t, \mathbf{x}_t} \mathbf{J}^{e\mathsf{T}}_{\mathbf{s}_t, \mathbf{x}_t} - \mathbf{J}^e_{\mathbf{s}_t, \mathbf{x}_t} \mathbf{J}^e_{\mathbf{s}_t, \mathbf{x}_t} \right] \mathbf{v} \\
&= \mathcal{L}^c_{QC} + \mathbb{E}_{\mathbf{v}, t, \mathbf{x}_t} \mathbf{v}^{\mathsf{T}} \left[ \mathbf{J}^e_{\mathbf{s}_t, \mathbf{x}_t} \mathbf{J}^{e\mathsf{T}}_{\mathbf{s}_t, \mathbf{x}_t} - \mathbf{J}^e_{\mathbf{s}_t, \mathbf{x}_t} \mathbf{J}^e_{\mathbf{s}_t, \mathbf{x}_t} \right] \mathbf{v}.
\end{aligned}
\tag{23}
$$

Because $\nabla_\phi \mathbf{v}^{\mathsf{T}} \left[ \mathbf{J}^e_{\mathbf{s}_t, \mathbf{x}_t} \mathbf{J}^{e\mathsf{T}}_{\mathbf{s}_t, \mathbf{x}_t} - \mathbf{J}^e_{\mathbf{s}_t, \mathbf{x}_t} \mathbf{J}^e_{\mathbf{s}_t, \mathbf{x}_t} \right] \mathbf{v} = \mathbf{0}$, therefore, we have

$$\nabla_\phi \mathcal{L}^{est}_{QC} = \nabla_\phi \mathcal{L}^c_{QC}. \tag{24}$$

## D.3 Proof of Theorem 4.1

**Theorem D.1** *Suppose the Frobenius norm of $\mathbf{J}^e_{\mathbf{s}_t, \mathbf{x}_t}$ is uniformly bounded by $M$, we have*

$$\mathcal{L}^c_{QC} \leq 2\sqrt{2}M \sqrt{\mathcal{L}^{simple}_{QC}} + \mathcal{L}^{simple}_{QC}, \tag{25}$$

*which indicates that if the simplied loss is zero, the original loss is also zero.*

$$
\begin{aligned}
\mathcal{L}_{QC}^{c} =&\mathbb{E}_{\mathbf{v},t,\mathbf{x_t}}\mathbf{v}^{\mathsf{T}}\left[2\mathbf{J}_{\mathbf{s}_t,\mathbf{x}_t}^{e}\mathbf{J}_{\mathbf{s}_t,\mathbf{x}_t}^{c\mathsf{T}} - 2\mathbf{J}_{\mathbf{s}_t,\mathbf{x}_t}^{e}\mathbf{J}_{\mathbf{s}_t,\mathbf{x}_t}^{c} + \mathbf{J}_{\mathbf{s}_t,\mathbf{x}_t}^{c}\mathbf{J}_{\mathbf{s}_t,\mathbf{x}_t}^{c\mathsf{T}} - \mathbf{J}_{\mathbf{s}_t,\mathbf{x}_t}^{c}\mathbf{J}_{\mathbf{s}_t,\mathbf{x}_t}^{c}\right]\mathbf{v} \\
&+ \mathbb{E}_{\mathbf{v},t,\mathbf{x_t}}\mathbf{v}^{\mathsf{T}}\left[\mathbf{J}_{\mathbf{s}_t,\mathbf{x}_t}^{e}\mathbf{J}_{\mathbf{s}_t,\mathbf{x}_t}^{e\mathsf{T}} - \mathbf{J}_{\mathbf{s}_t,\mathbf{x}_t}^{e}\mathbf{J}_{\mathbf{s}_t,\mathbf{x}_t}^{e}\right]\mathbf{v} \\
&- \mathbb{E}_{\mathbf{v},t,\mathbf{x_t}}\mathbf{v}^{\mathsf{T}}\left[\mathbf{J}_{\mathbf{s}_t,\mathbf{x}_t}^{e}\mathbf{J}_{\mathbf{s}_t,\mathbf{x}_t}^{e\mathsf{T}} - \mathbf{J}_{\mathbf{s}_t,\mathbf{x}_t}^{e}\mathbf{J}_{\mathbf{s}_t,\mathbf{x}_t}^{e}\right]\mathbf{v} \\
=&\mathbb{E}_{\mathbf{v},t,\mathbf{x_t}}\mathbf{v}^{\mathsf{T}}\left[\left(\mathbf{J}_{\mathbf{s}_t,\mathbf{x}_t}^{e} + \mathbf{J}_{\mathbf{s}_t,\mathbf{x}_t}^{c}\right)\left(\mathbf{J}_{\mathbf{s}_t,\mathbf{x}_t}^{e} + \mathbf{J}_{\mathbf{s}_t,\mathbf{x}_t}^{c}\right)^{\mathsf{T}} - \left(\mathbf{J}_{\mathbf{s}_t,\mathbf{x}_t}^{e} + \mathbf{J}_{\mathbf{s}_t,\mathbf{x}_t}^{c}\right)\left(\mathbf{J}_{\mathbf{s}_t,\mathbf{x}_t}^{e} + \mathbf{J}_{\mathbf{s}_t,\mathbf{x}_t}^{c}\right)\right]\mathbf{v} \\
&- \mathbb{E}_{\mathbf{v},t,\mathbf{x_t}}\mathbf{v}^{\mathsf{T}}\left[\mathbf{J}_{\mathbf{s}_t,\mathbf{x}_t}^{e}\mathbf{J}_{\mathbf{s}_t,\mathbf{x}_t}^{e\mathsf{T}} - \mathbf{J}_{\mathbf{s}_t,\mathbf{x}_t}^{e}\mathbf{J}_{\mathbf{s}_t,\mathbf{x}_t}^{e}\right]\mathbf{v} \\
=&\mathbb{E}_{t,\mathbf{x_t}}\left[\operatorname{tr}\left(\left(\mathbf{J}_{\mathbf{s}_t,\mathbf{x}_t}^{e} + \mathbf{J}_{\mathbf{s}_t,\mathbf{x}_t}^{c}\right)\left(\mathbf{J}_{\mathbf{s}_t,\mathbf{x}_t}^{e} + \mathbf{J}_{\mathbf{s}_t,\mathbf{x}_t}^{c}\right)^{\mathsf{T}}\right) - \operatorname{tr}\left(\left(\mathbf{J}_{\mathbf{s}_t,\mathbf{x}_t}^{e} + \mathbf{J}_{\mathbf{s}_t,\mathbf{x}_t}^{c}\right)\left(\mathbf{J}_{\mathbf{s}_t,\mathbf{x}_t}^{e} + \mathbf{J}_{\mathbf{s}_t,\mathbf{x}_t}^{c}\right)\right)\right] \\
&- \mathbb{E}_{t,\mathbf{x_t}}\left[\operatorname{tr}\left(\mathbf{J}_{\mathbf{s}_t,\mathbf{x}_t}^{e}\mathbf{J}_{\mathbf{s}_t,\mathbf{x}_t}^{e\mathsf{T}}\right) - \operatorname{tr}\left(\mathbf{J}_{\mathbf{s}_t,\mathbf{x}_t}^{e}\mathbf{J}_{\mathbf{s}_t,\mathbf{x}_t}^{e}\right)\right] \\
=&\frac{1}{2}\mathbb{E}_{t,\mathbf{x_t}}\left\|\left(\mathbf{J}_{\mathbf{s}_t,\mathbf{x}_t}^{e} + \mathbf{J}_{\mathbf{s}_t,\mathbf{x}_t}^{c}\right) - \left(\mathbf{J}_{\mathbf{s}_t,\mathbf{x}_t}^{e} + \mathbf{J}_{\mathbf{s}_t,\mathbf{x}_t}^{c}\right)^{\mathsf{T}}\right\|_{F}^{2} - \frac{1}{2}\mathbb{E}_{t,\mathbf{x_t}}\left\|\mathbf{J}_{\mathbf{s}_t,\mathbf{x}_t}^{e} - \mathbf{J}_{\mathbf{s}_t,\mathbf{x}_t}^{e\mathsf{T}}\right\|_{F}^{2} \\
=&\frac{1}{2}\mathbb{E}_{t,\mathbf{x_t}}\left\|\left(\mathbf{J}_{\mathbf{s}_t,\mathbf{x}_t}^{e} - \mathbf{J}_{\mathbf{s}_t,\mathbf{x}_t}^{e\mathsf{T}}\right) + \left(\mathbf{J}_{\mathbf{s}_t,\mathbf{x}_t}^{c} - \mathbf{J}_{\mathbf{s}_t,\mathbf{x}_t}^{c\mathsf{T}}\right)\right\|_{F}^{2} - \frac{1}{2}\mathbb{E}_{t,\mathbf{x_t}}\left\|\mathbf{J}_{\mathbf{s}_t,\mathbf{x}_t}^{e} - \mathbf{J}_{\mathbf{s}_t,\mathbf{x}_t}^{e\mathsf{T}}\right\|_{F}^{2} \\
\leq&\frac{1}{2}\mathbb{E}_{t,\mathbf{x_t}}\left(\left\|\mathbf{J}_{\mathbf{s}_t,\mathbf{x}_t}^{e} - \mathbf{J}_{\mathbf{s}_t,\mathbf{x}_t}^{e\mathsf{T}}\right\|_{F} + \left\|\mathbf{J}_{\mathbf{s}_t,\mathbf{x}_t}^{c} - \mathbf{J}_{\mathbf{s}_t,\mathbf{x}_t}^{c\mathsf{T}}\right\|_{F}\right)^{2} - \frac{1}{2}\mathbb{E}_{t,\mathbf{x_t}}\left\|\mathbf{J}_{\mathbf{s}_t,\mathbf{x}_t}^{e} - \mathbf{J}_{\mathbf{s}_t,\mathbf{x}_t}^{e\mathsf{T}}\right\|_{F}^{2} \\
=&\mathbb{E}_{t,\mathbf{x_t}}\left\|\mathbf{J}_{\mathbf{s}_t,\mathbf{x}_t}^{e} - \mathbf{J}_{\mathbf{s}_t,\mathbf{x}_t}^{e\mathsf{T}}\right\|_{F}\left\|\mathbf{J}_{\mathbf{s}_t,\mathbf{x}_t}^{c} - \mathbf{J}_{\mathbf{s}_t,\mathbf{x}_t}^{c\mathsf{T}}\right\|_{F} + \frac{1}{2}\mathbb{E}_{t,\mathbf{x_t}}\left\|\mathbf{J}_{\mathbf{s}_t,\mathbf{x}_t}^{c} - \mathbf{J}_{\mathbf{s}_t,\mathbf{x}_t}^{c\mathsf{T}}\right\|_{F}^{2}
\end{aligned}
\tag{26}
$$

Because that $\left\|\mathbf{J}_{\mathbf{s}_t,\mathbf{x}_t}^{e}\right\|_{F} \leq M$, we have:

$$
\left\|\mathbf{J}_{\mathbf{s}_t,\mathbf{x}_t}^{c} - \mathbf{J}_{\mathbf{s}_t,\mathbf{x}_t}^{c\mathsf{T}}\right\|_{F} \leq \left\|\mathbf{J}_{\mathbf{s}_t,\mathbf{x}_t}^{c}\right\|_{F} + \left\|\mathbf{J}_{\mathbf{s}_t,\mathbf{x}_t}^{c\mathsf{T}}\right\|_{F} \leq 2M.
\tag{27}
$$

Then, we have

$$
\mathcal{L}_{QC}^{c} \leq 2M\mathbb{E}_{t,\mathbf{x_t}}\left\|\mathbf{J}_{\mathbf{s}_t,\mathbf{x}_t}^{e} - \mathbf{J}_{\mathbf{s}_t,\mathbf{x}_t}^{e\mathsf{T}}\right\|_{F} + \frac{1}{2}\mathbb{E}_{t,\mathbf{x_t}}\left\|\mathbf{J}_{\mathbf{s}_t,\mathbf{x}_t}^{c} - \mathbf{J}_{\mathbf{s}_t,\mathbf{x}_t}^{c\mathsf{T}}\right\|_{F}^{2}.
\tag{28}
$$

By Cauchy-Schwarz Inequality, we have

$$
\left[\mathbb{E}_{t,\mathbf{x_t}}\left\|\mathbf{J}_{\mathbf{s}_t,\mathbf{x}_t}^{e} - \mathbf{J}_{\mathbf{s}_t,\mathbf{x}_t}^{e\mathsf{T}}\right\|_{F}\right]^{2} \leq \mathbb{E}_{t,\mathbf{x_t}}\left\|\mathbf{J}_{\mathbf{s}_t,\mathbf{x}_t}^{c} - \mathbf{J}_{\mathbf{s}_t,\mathbf{x}_t}^{c\mathsf{T}}\right\|_{F}^{2}.
\tag{29}
$$

Therefore, we have

$$
\mathbb{E}_{t,\mathbf{x_t}}\left\|\mathbf{J}_{\mathbf{s}_t,\mathbf{x}_t}^{e} - \mathbf{J}_{\mathbf{s}_t,\mathbf{x}_t}^{e\mathsf{T}}\right\|_{F} \leq \sqrt{\mathbb{E}_{t,\mathbf{x_t}}\left\|\mathbf{J}_{\mathbf{s}_t,\mathbf{x}_t}^{c} - \mathbf{J}_{\mathbf{s}_t,\mathbf{x}_t}^{c\mathsf{T}}\right\|_{F}^{2}}.
\tag{30}
$$

Then,

$$
\mathcal{L}_{QC}^{c} \leq 2\sqrt{2}M\sqrt{\frac{1}{2}\mathbb{E}_{t,\mathbf{x_t}}\left\|\mathbf{J}_{\mathbf{s}_t,\mathbf{x}_t}^{c} - \mathbf{J}_{\mathbf{s}_t,\mathbf{x}_t}^{c\mathsf{T}}\right\|_{F}^{2}} + \frac{1}{2}\mathbb{E}_{t,\mathbf{x_t}}\left\|\mathbf{J}_{\mathbf{s}_t,\mathbf{x}_t}^{c} - \mathbf{J}_{\mathbf{s}_t,\mathbf{x}_t}^{c\mathsf{T}}\right\|_{F}^{2},
\tag{31}
$$

which indicates

$$
\mathcal{L}_{QC}^{c} \leq 2\sqrt{2}M\sqrt{\mathcal{L}_{QC}^{simple}} + \mathcal{L}_{QC}^{simple}.
\tag{32}
$$

# E    MGDA FOR FEATURE INJECTION AND COMBINATION

The score function $\mathbf{s}_t(\mathbf{x}_t)$ is defined as the gradient of the scalar value $\log p(\mathbf{x}_t)$, we can interpret addition operation in the score domain as the combination of gradients from different optimization objectives. Thus, MGDA is applicable for optimizing this blend of diverse gradients. While the feature domain of U-Net may not present a straightforward optimization objective, we adapt the

principle of forming acute angles between each feature map to mitigate conflicts among various features.

The distinction between feature injection and combination lies in the architecture of ControlNet. Feature injection involves adding the control signal to the original U-Net feature map, which suggests that the original U-Net feature map should ideally remain unchanged. Therefore, after balancing the coefficients with MGDA, additional scaling is required to ensure this. In contrast, for feature combinations, we can directly apply MGDA to balance the feature maps without such constraints.

# F    IMPLEMENTATION DETAILS ABOUT MICONTROLNET

## F.1    DATA REBALANCE DETAILS

Firstly, we segment the images into distinct regions. Subsequently, we randomly select portions of these segmentations, utilizing both the edge-like features within these selected areas and the original images to construct our training dataset. This approach ensures that edge-like features not included in the segmentations are converted into silent control signals. Consequently, the corresponding image regions retain high-frequency information, crucial for detailed image generation in silent control signals.

## F.2    TRAINING DETAILS

Our training process comprises two stages, all of which are conducted on the MultiGen-20M dataset (Qin et al., 2023) using our balanced control signals. In the first stage, we train the model using the $\mathbf{add}^{inj}$ operation for 2 epochs. For the OpenPose Model, which has less training data, the duration extends to 9 epochs. In the subsequent stage, we integrate the $\mathcal{L}_{QC}^{simple}$ loss into the original diffusion predicting noise loss with a coefficient of 0.01, and continue training for 2000 steps with an equivalent batch size of 128. All experiments are executed on eight NVIDIA A800 GPUs, each with 80GB of memory. The first stage requires approximately 2 days, while the second stage is completed in about 7 hours.

## F.3    SAMPLING DETAILS

For sampling with multi MIControlNets, we first apply the $\mathbf{add}^{com}$ operation for the feature combination of different MIControlNets. Then we apply the $\mathbf{add}^{inj}$ operation to add the feature maps of MIControlNets to that of the original U-Net.

# G    MORE EXPERIMENTS

## G.1    THE SUDDEN CONVERGENCE OF MICONTROLNET AND FID

We evaluate the rapid convergence behavior of MIControlNet compared to the original ControlNet, as illustrated in Figure 6. Notably, ControlNet often experiences sudden shifts in performance at particular training steps. To investigate further, we focused on these critical training milestones for both MIControlNet and ControlNet. Our results demonstrate that MIControlNet achieves earlier convergence while maintaining similar or improved generation quality compared to ControlNet.

## G.2    MORE QUALITATIVE METRICS FOR MULTI-CONDITION EVALUATION

Table 2 shows the $L1$ distance between the extracted condition from the generated images and the original condition. Our MIControlNet achieves the lowest values in most cases, indicating better preservation of control signals (excluding silent control signals) in the generated images.

Table 3 presents the FID scores of various models using a new conditioning approach, where the ground truth image is split into left and right sections, and conditions are extracted for each part. In Table 4, the ground truth image is divided into the central object and the surrounding areas, with conditions extracted accordingly. Our MIControlNet consistently suppresses baseline models across nearly all conditions, demonstrating its superior performance in multi-condition image generation.

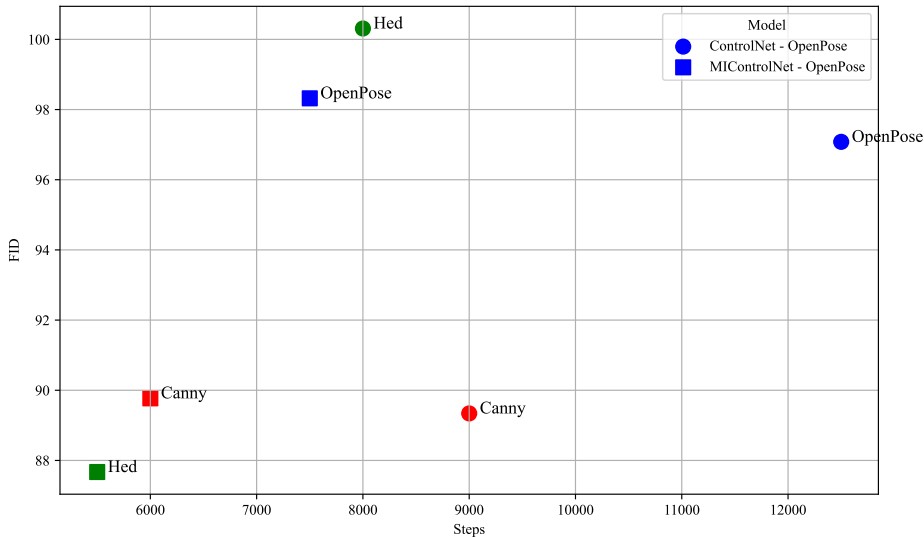

Figure 6: FID-convergence steps. circle represents ControlNet and square represents MIControlNet.

Table 2: The distance between the condition extracted from the generated image and the ground truth in the conflict area. The distances are $L1$ norm expressed in units of $1 \times 10^4$.

| Methods | Openpose-Canny | Openpose-Hed | Canny-Hed |
|---|---|---|---|
| ControlNet | 1.3903 | 1.7851 | 2.8626 |
| ControlNet$^{0.5}$ | 1.3223 | 1.8310 | 2.8881 |
| ControlNet$^{1.5}$ | 1.3848 | 1.9009 | 2.8123 |
| ControlNet$^*$ | 1.3833 | 1.9066 | 2.9381 |
| Ours (1-stage) | **0.9638** | **1.5080** | **1.9634** |
| Ours (2-stage) | **1.0729** | **1.6600** | 2.1954 |
| Uni-ControlNet | 1.0808 | 1.7232 | **2.0951** |

### G.3 QUALITATIVE METRICS UNDER DIFFERENT PROMPT CONDITIONS

Table 5 presents the FID and Total Variance (TV, in units of $1 \times 10^4$) for Canny and OpenPose conditions under three scenarios: no prompts, brief prompts, and detailed prompts.

We have the following findings:

- MIControlNet, with or without the conservativity loss, demonstrates similar FID performance. However, with conservativity loss, MIControlNet exhibits improved pattern generation ability under silent control signals, as highlighted in Table 5.

- MIControlNet achieves comparable FID performance to the baseline but demonstrates significantly stronger performance in terms of total variance.

- When comparing no prompts, brief prompts, and detailed prompts, providing more detailed prompts generally leads to better FID performance and smaller total variance.

- Interestingly, for detailed prompts, the total variance tends to slightly increase. We hypothesize that this is due to the more detailed prompts offering finer control under silent control signals, thereby generating more diverse patterns.

Table 3: The FIDs for the left-right split condition. ControlNet*** denotes ControlNet** with our balanced feature combination sampling. ControlNet**$^{0.5}$ and ControlNet**$^{1.5}$ represent ControlNet** sampling where the first control signal is scaled by 0.5 and 1.5, respectively.

| Methods | Openpose-Canny | Openpose-Hed | Canny-Hed | HED-Depth |
|---|---|---|---|---|
| ControlNet | **63.8590** | 75.2076 | 58.8519 | 69.3344 |
| ControlNet** | 68.7007 | 69.4012 | 61.9949 | 62.1554 |
| ControlNet*** | 67.5028 | 65.9667 | 66.9362 | 62.9284 |
| ControlNet**$^{0.5}$ | 80.1214 | 84.4420 | 68.6729 | 61.8351 |
| ControlNet**$^{1.5}$ | 66.1629 | **65.2359** | 65.5698 | 71.7198 |
| ControlNet* | 68.8815 | 68.0828 | 56.7660 | 71.9574 |
| ControlNet$^{0.5}$ | 65.9972 | 108.2335 | 81.5698 | 75.4173 |
| ControlNet$^{1.5}$ | 69.2258 | 66.4017 | 58.4945 | 93.3087 |
| Ours(1-stage) | 64.7937 | **64.0063** | **55.9595** | **56.3233** |
| Ours(2-stage) | **62.5830** | 66.4729 | **54.3970** | **57.9421** |
| Uni-ControlNet | 71.2586 | 89.4048 | 56.8694 | 65.9861 |

Table 4: The FIDs for the central-outside split condition.

| Methods | Openpose-Canny | Openpose-Hed | Canny-Hed | Hed-Depth |
|---|---|---|---|---|
| ControlNet | 65.2961 | 75.3146 | 60.8962 | 73.4688 |
| ControlNet** | 68.1883 | 63.8602 | 58.9864 | 59.8399 |
| ControlNet*** | 66.6215 | 66.4260 | 62.1626 | 62.1199 |
| ControlNet**$^{0.5}$ | 70.8668 | 71.1370 | 63.1586 | 61.1704 |
| ControlNet**$^{1.5}$ | 67.3933 | 65.0094 | 61.3308 | 62.3237 |
| ControlNet* | 72.1651 | 72.1254 | 67.7424 | 78.5825 |
| ControlNet$^{0.5}$ | 72.5424 | 104.6354 | 82.8161 | 81.8447 |
| ControlNet$^{1.5}$ | 67.6942 | 67.9931 | 62.5344 | 89.5242 |
| Ours(1-stage) | 62.9185 | **59.0101** | **54.9762** | **56.8017** |
| Ours(2-stage) | **61.6007** | **61.2391** | **56.3576** | **57.4142** |
| Uni-ControlNet | **60.5932** | 68.3847 | 57.8282 | 61.7094 |

## G.4 ASYMMETRY ANALYSIS FOR REGULAR CONTROLNET

We calculate the asymmetry (*Asym*) for Regular ControlNet and compare it with our MIControlNet (1-stage) and MIControlNet (2-stage). The results are shown in Table 6:

We have the following findings:

- MIControlNet (1-stage) performs slightly better than ControlNet in terms of asymmetry (*Asym*).

- MIControlNet (2-stage) significantly outperforms both ControlNet and MIControlNet (1-stage) on *Asym*.

- For each condition, MIControlNet (2-stage) demonstrates consistent improvements on a logarithmic scale.

## G.5 A MORE CLEAR ABLATION

The FID scores for a thorough ablation study are shown in Table 7. We observe that:

- Our silent control signal-targeted data augmentation, feature injection & combination, and conservativity loss all lead to improvements in FID scores.

- The conservativity loss, particularly for Canny combined with other conditions, achieves a consistent improvement of approximately 3 points in FID.

Table 5: FID and Total Variance (TV) for Canny and OpenPose conditions under different prompt scenarios.

| (FID, TV) | ControlNet | MIControlNet (1-stage) | MIControlNet (2-stage) |
|---|---|---|---|
| Canny No Prompts | (109.6, 2.62) | (114.4, 3.54) | (123.9, 3.79) |
| Canny Brief Prompts | (89.34, 2.48) | (89.77, 3.24) | (90.18, 3.15) |
| Canny Detailed Prompts | (88.55, 2.47) | (90.21, 3.28) | (**89.37**, 3.36) |
| OpenPose No Prompts | (132.5, 3.34) | (131.9, 3.39) | (133.0, 3.67) |
| OpenPose Brief Prompts | (97.08, 2.52) | (98.32, 2.71) | (98.14, 2.74) |
| OpenPose Detailed Prompts | (99.09, 2.70) | (95.34, 2.92) | (**94.16**, 2.91) |

Table 6: Asymmetry (*Asym*) comparison across different conditions.

| Condition | Canny | Hed | Openpose |
|---|---|---|---|
| ControlNet | 56.75 | 22.41 | 6.454 |
| MIControlNet (1-stage) | 29.87 | 38.28 | 3.980 |
| MIControlNet (2-stage) | 0.1174 | 0.1894 | 0.0274 |

- The improvements achieved through the conservativity loss are consistent, and we have further strengthened its theoretical foundation, particularly in the context of modular neural networks designed to optimize GPU memory usage and computational efficiency.

Table 7: FID scores for different methods under various conditions. Lower scores indicate better performance.

| Method | Openpose-Canny | Canny-Hed | Hed-Depth |
|---|---|---|---|
| Vanilla ControlNet | 80.37 / 111.30 | 123.59 / 86.43 | 91.98 / 86.25 |
| + Data Augmentation | 92.98 / 84.02 | 77.02 / 75.46 | 74.28 / 81.16 |
| + Our Feature Injection & Combination | 76.13 / 77.22 | 74.19 / 70.26 | 71.16 / 71.93 |
| + Conservativity Loss | **75.77 / 72.25** | **71.34 / 69.35** | **69.68 / 71.18** |

## H   SAMPLES

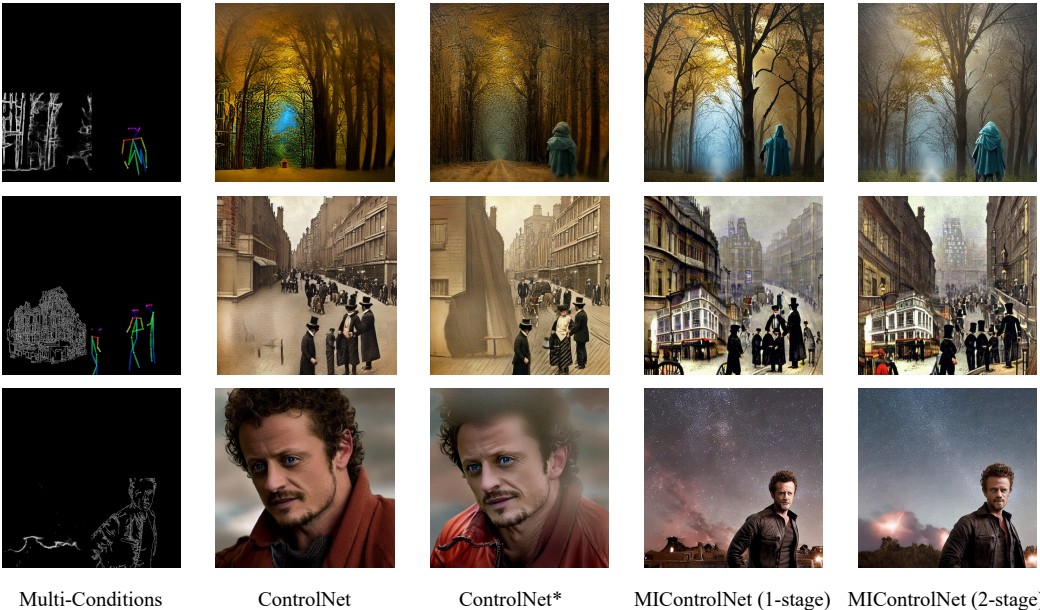

Multi-Conditions    ControlNet    ControlNet*    MIControlNet (1-stage)    MIControlNet (2-stage)

Figure 7: Comparison of ControlNet and MIControlNet for multi-conditions generation.

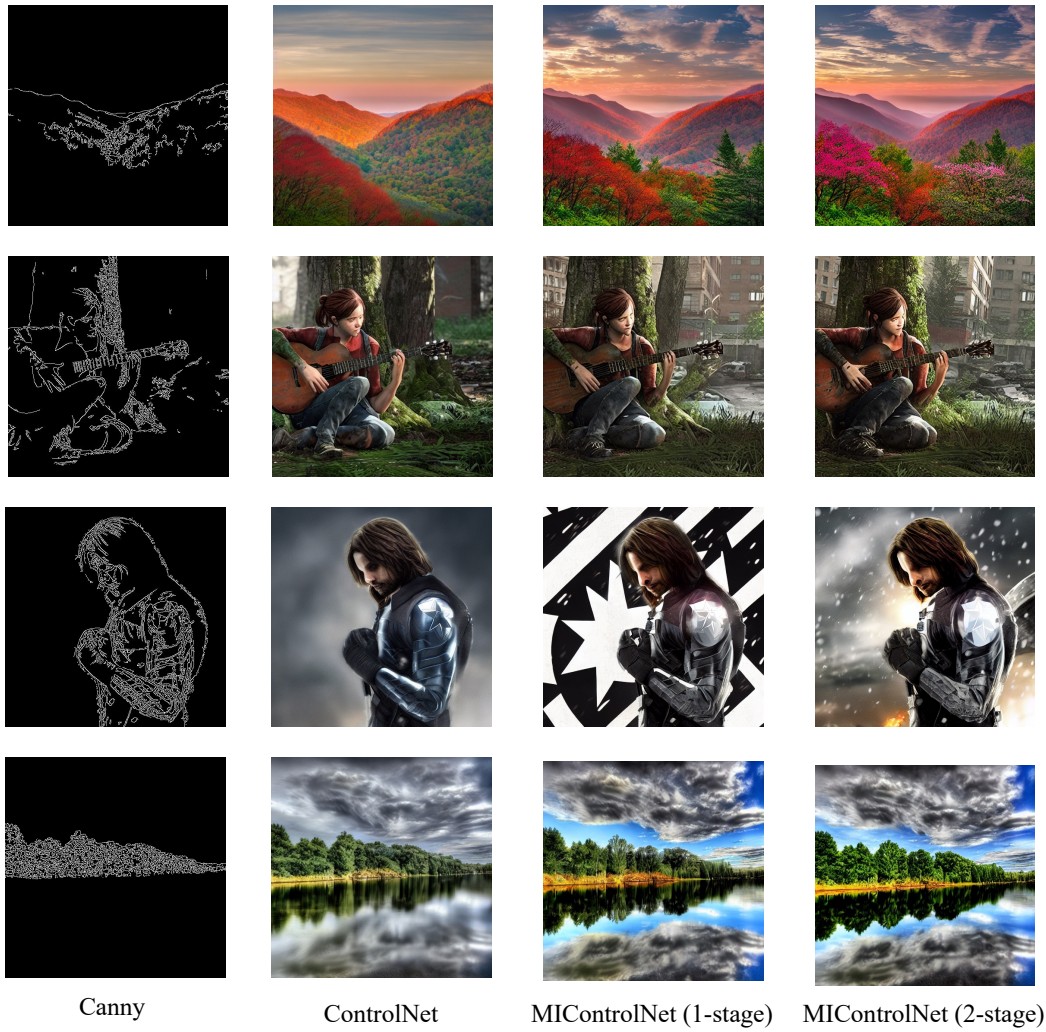

Canny  ControlNet  MIControlNet (1-stage)  MIControlNet (2-stage)

Figure 8: More visual results for single control signal.

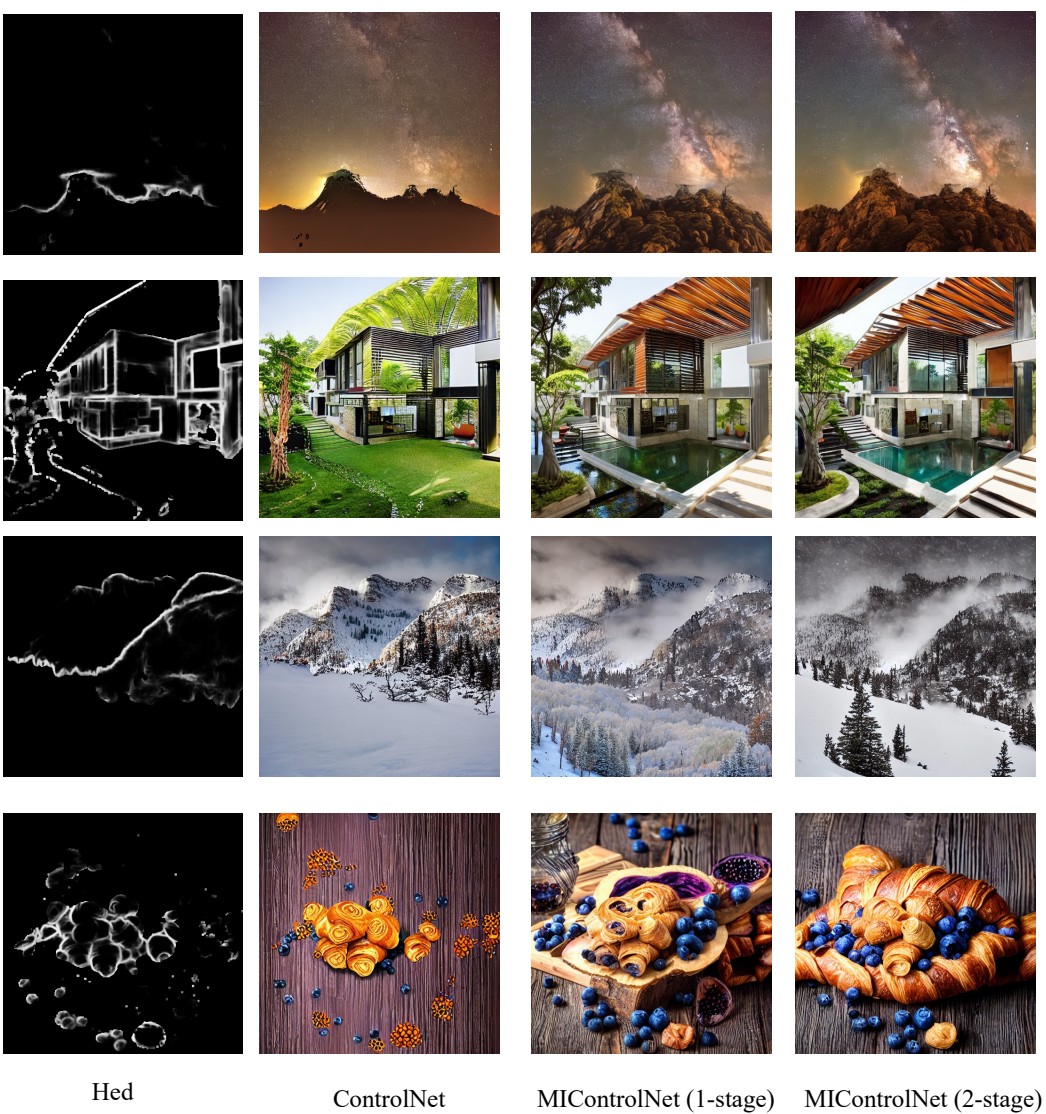

Hed        ControlNet        MIControlNet (1-stage)    MIControlNet (2-stage)

Figure 9: More visual results for single control signal.

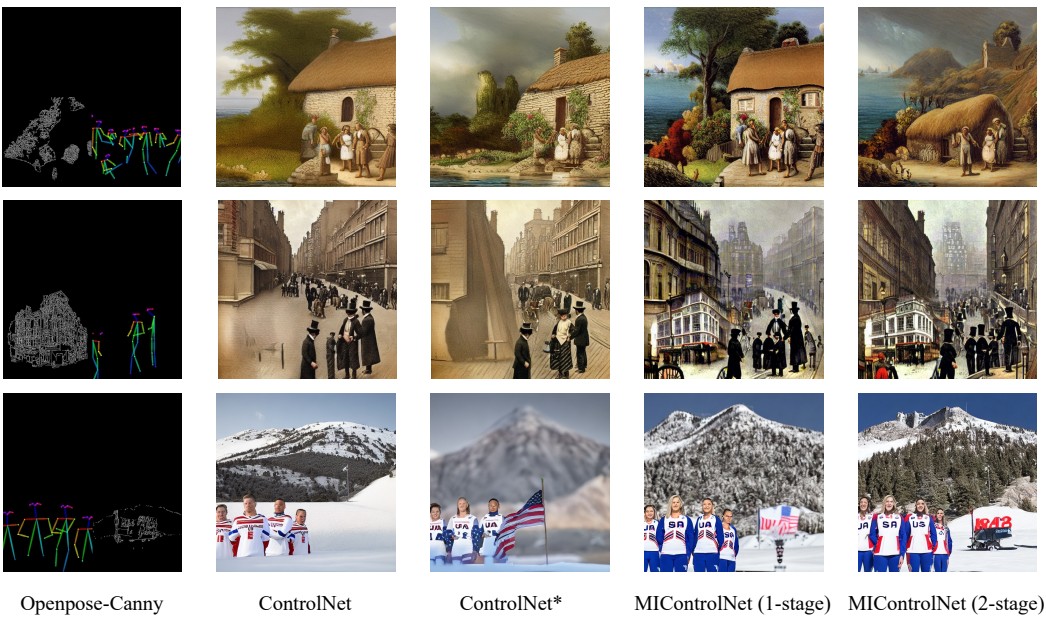

|  | Openpose-Canny | ControlNet | ControlNet* | MIControlNet (1-stage) | MIControlNet (2-stage) |

Figure 10: More visual results for multi-control signals.

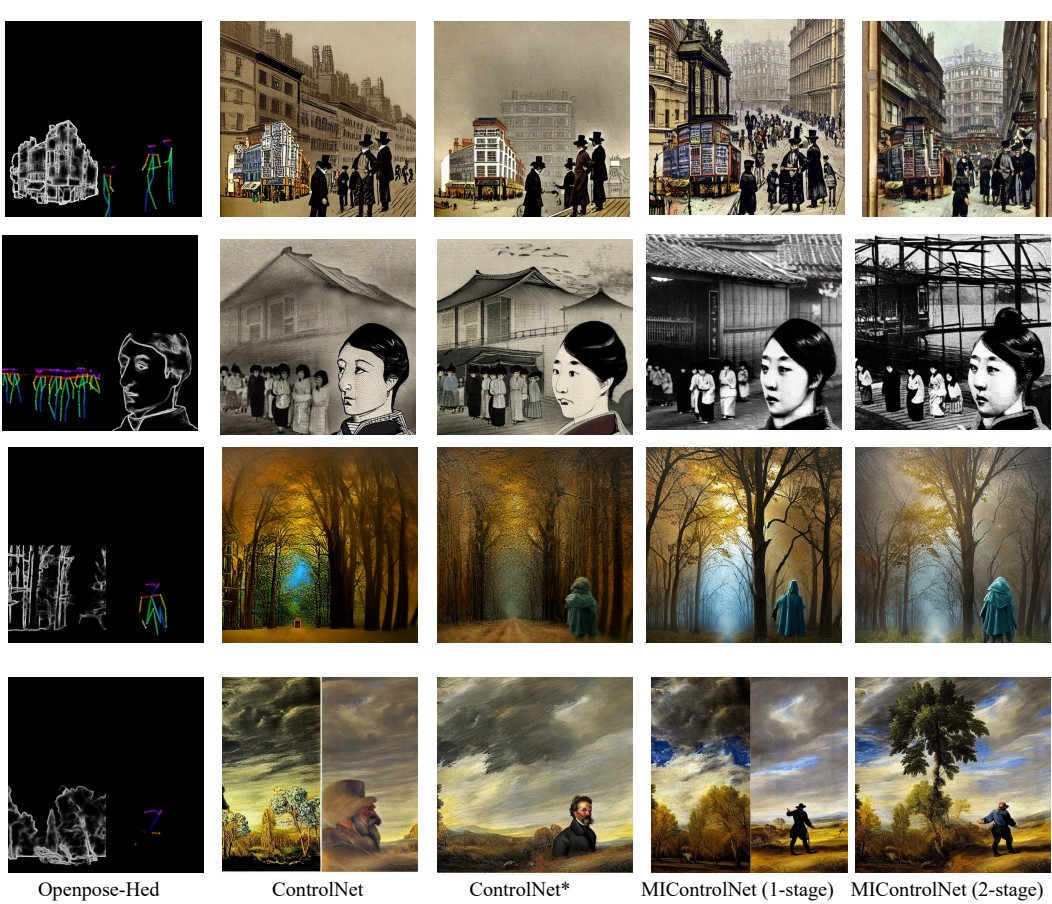

|  | Openpose-Hed | ControlNet | ControlNet* | MIControlNet (1-stage) | MIControlNet (2-stage) |

Figure 11: More visual results for multi-control signals.

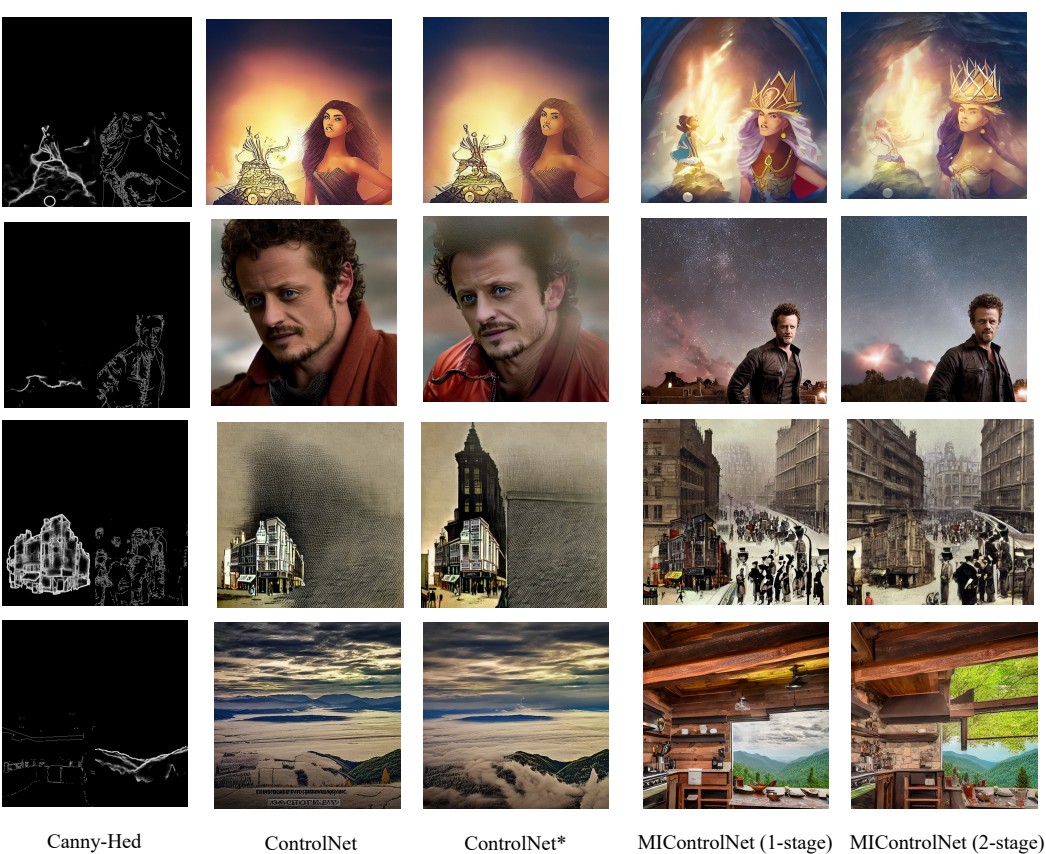

Canny-Hed  ControlNet  ControlNet*  MIControlNet (1-stage)  MIControlNet (2-stage)

Figure 12: More visual results for multi-control signals.

