# OpenReview forum: "Minimal Impact ControlNet: Advancing Multi-ControlNet Integration"
_ICLR.cc/2025/Conference — ICLR 2025 Poster_

### Official Review · Reviewer_TnUC · 2024-10-29

**Soundness:** 2
**Presentation:** 1
**Contribution:** 2
**Rating:** 6
**Confidence:** 3

**Summary:**

This article addresses the coordination of diverse control signals in diffusion models for precise manipulation, specifically focusing on the conflicts caused by low-frequency signals, termed "silent control signals." Such signals often interfere with those that generate finer details, resulting in suboptimal outcomes. To tackle this, the authors propose the Minimal Impact ControlNet (MIControlNet) framework, which minimizes ControlNet’s influence on the original U-Net and mitigates conflicts through modified training data, multi-objective optimization, and adjustments to the Jacobian matrix asymmetry in the score function. A two-stage training process is used: first, balanced data and feature injection for various conditions, followed by a secondary phase incorporating a conservativity loss to enhance signal compatibility and image fidelity.

**Strengths:**

1. Analyzed challenges in training diffusion models with multiple control signals for ControlNet:

* Data Bias: Handling biases in regions influenced by "silent" control signals.

* Optimal Combination Ratios: Determining ideal ratios for integrating multiple ControlNet layers.

* Conditional Score Function Stability: Ensuring balanced, stable updates across control signals (referred to as the “conservativity” of the score function).

2. Introduced a conservativity loss function within a modular network structure, supported by theoretical analysis.
3. Results demonstrate that the new loss function improves texture details specifically in areas affected by silent control signals, achieving finer and more accurate generation.

**Weaknesses:**

1. ControlNet is not yet a widely recognized concept. It is recommended that the authors briefly introduce its definition when first mentioned; otherwise, readers may feel confused starting from the abstract.

2. Some phrases lack clarity. For example, in lines 18-19 of the abstract, "..., the silent control signals can suppress the generation of textures in related areas,..." leaves "related areas" ambiguous. Does this refer to areas that are texturally similar, spatially close, or something else?

3. There is a lack of necessary citations, such as in lines 164-165, where it states "to find the Pareto optimal solution, a state where no single objective can be improved...". Adding citations here would provide more credibility to the explanation.

4. The organization of the paper could be improved, as it is currently somewhat challenging to follow. Additionally, the main text does not sufficiently emphasize results showing the balancing of coefficients among multiple control signals.

5. The three main issues highlighted in the problem analysis section lack clear support in the experimental results.

6. Ablation studies are missing, which would help illustrate the individual contributions of each component to the overall outcome.

**Questions:**

* In Equations (8) and (9), what do v1 and v2 represent? Additionally, since λi varies across layers, should v1 and v2 also include subscripts to indicate properties at different layers within the network?

* The concept of "conservativity" in ControlNet doesn’t seem to be a commonly used term. I find it somewhat confusing—could the authors provide further clarification?
* In Figure 4(b), why is there no comparison with ControlNet? From the results, it appears that Stage-1 has a more significant impact than Stage-2. How do the authors view the balance between these two stages? This may relate to the previous question, as the authors mention that Stage-2 primarily enhances the "conservativity" of control signals.

* Could this approach potentially be extended by first separating foreground and background for guided generation and then synthesizing the layers? This might allow for even greater control over specific elements.

---

> ### Author Response · Authors · 2024-11-23
>
> We extend our heartfelt gratitude to reviewer TnUC for recognizing the depth of our problem analysis and the theoretical contributions of our work. Your detailed suggestions helped us include additional details in the preliminaries and provide a clearer explanation of the conservativity loss, enabling our work to reach and resonate with a broader audience.
> ## W1-3: About writing.
> * For ControlNet[1], we will reference the work earlier in the text and provide a brief introduction to its definition.
> * We apologize for the lack of clarity. Because the signal has a shape like $C \times H \times W$, and the silent control signal lies in the $H \times W$ portion, the term “related areas” refers to the region within $H \times W$ where the silent control signal is located.
> * Sorry for the lack of citations about Pareto optimal solution, We will add the related work[2-3] in the final version.
>
> ## W4: More analysis on the balancing of coefficients among multiple control signals.
>
> For our methods, the combination of different features varies across timesteps and layers, making it challenging to analyze systematically. However, we compare our method with different fixed coefficients. We introduce fixed coefficients for the first control signals, labeled ControlNet$^{0.5}$ and ControlNet$^{1.5}$, maintaining a total scaling factor of 2.0 to match the original ControlNet and our feature combination method. The FID performance is shown in the following table.
>
> | Methods            | Openpose-Canny    | Openpose-Hed          | Canny-Hed             |
> | ------------------ | ----------------- | --------------------- | --------------------- |
> | ControlNet         | 80.37 / 111.30    | 76.98 / 84.20         | 123.59 / 86.43        |
> | ControlNet$^{0.5}$ | 105.86 / 123.13   | 145.88 / 107.52       | 143.67 / 106.40       |
> | ControlNet$^{1.5}$ | **74.37** / 99.44 | 74.52 / 86.57         | 120.84 / 88.38        |
> | Ours (1-stage)     | 76.13 / **77.22** | **70.32** / **68.42** | **74.19** / **70.26** |
>
> We observe that our dynamic balanced coefficients generally outperform fixed coefficients, demonstrating the effectiveness of balancing of coefficients among multiple control signals.
> ## W5-6: A more detailed ablation.
>
> We apologize for not providing a clear ablation analysis of our three contributions earlier. To address this, we first added the $Asym$ component for the baselines in Figure 4.(b) as part of the second experiment in **Common Concern 1**. Furthermore, we have included a comprehensive ablation study, as detailed in **Common Concern 2**.
> ## Q1: Symbols in Equation (8) and (9).
> We apologize for the lack of clarity. Here, $v_1$ and $v_2$ are simply vectors of the same shape. What we aim to introduce is the operator symbol $\lambda_i$. For each layer, $\lambda_i(\cdot, \cdot)$ takes the features of the corresponding layer as input. Consequently, the properties at different layers naturally differ due to the layer-specific inputs and operations.
> ## Q2: Further clarification about conservativity.
> We apologize for not making this clear.
>
> **TL;DR:** If a vector field is the gradient of a scalar function, it exhibits the property of conservativity. For example, in real space, the gravitational field is the gradient of the gravitational potential energy, which makes the gravitational field conservative.
>
> In the context of Diffusion Models, the score function that the neural network aims to fit is expressed as $\nabla_{\mathbf{x}_t}\log p_t(\mathbf{x}_t)$. This gradient-of-a-scalar-function formulation inherently guarantees that the score function exhibits the conservativity property.

---

> ### Author Response · Authors · 2024-11-23
>
> ## Q3: Balance between the first stage and the second stage.
> We apologize for not including the related experiment initially. To address this, we have added the $Asym$ component to the baselines in Figure 4.(b) as part of the second experiment in **Common Concern 1**.
>
> In our experiments, training Stage-2 alongside Stage-1 proved challenging, likely due to the higher variance of the Stage-2 loss as estimated by Hutchinson’s trace estimator [4]. To address this issue, we adopted a post-training approach for Stage-2, restricting it to 2000 steps. Experimental results show that this shorter training duration is sufficient for the loss to effectively achieve its objective and consistently enhance FID performance across different tasks.
> ## Q4: Extend our methods by first separating foreground and background for guided generation and then synthesizing the layers.
>
> This is a highly promising idea. With the rapid advancements in base models, recent work [5] has shown the capability to directly generate images layer by layer. In such scenarios, challenges similar to those addressed in our problem settings are likely to emerge. We believe that applying our method to these new approaches offers a compelling direction for future research.
>
> [1] Zhang, Lvmin, Anyi Rao, and Maneesh Agrawala. "Adding conditional control to text-to-image diffusion models." _Proceedings of the IEEE/CVF International Conference on Computer Vision_. 2023.
>
> [2] Sener, Ozan, and Vladlen Koltun. "Multi-task learning as multi-objective optimization." _Advances in neural information processing systems_ 31 (2018).
>
> [3]Désidéri, Jean-Antoine. "Multiple-gradient descent algorithm (MGDA) for multiobjective optimization." _Comptes Rendus Mathematique_ 350.5-6 (2012): 313-318.
>
> [4] Hutchinson, Michael F. "A stochastic estimator of the trace of the influence matrix for Laplacian smoothing splines." _Communications in Statistics-Simulation and Computation_18.3 (1989): 1059-1076._
>
> [5]Zhang, Lvmin, and Maneesh Agrawala. "Transparent image layer diffusion using latent transparency." _arXiv preprint arXiv:2402.17113_ (2024).

---

> ### Author Response · Authors · 2024-11-28
>
> Dear Reviewer TnUC,
>
>
> Thank you for your thoughtful and insightful feedback. We have carefully addressed your concerns, including adding more details about **ControlNet and conservativity loss**, refining the manuscript, and **conducting more comprehensive ablation studies** to support the three key contributions.
>
>
> As the discussion phase draws to a close, we welcome any additional questions or suggestions you might have. We are happy to engage further and address any remaining points of concern.

---

> ### Author Response · Authors · 2024-12-03
>
> Dear Reviewer TnUC,
>
> As the discussion stage approaches its conclusion, we would like to inquire if you have any additional questions or suggestions. We would be delighted to engage in further discussion with you.

---

### Official Review · Reviewer_DjNC · 2024-11-01

**Soundness:** 4
**Presentation:** 3
**Contribution:** 3
**Rating:** 6
**Confidence:** 2

**Summary:**

In current ControlNet training, each control is designed to influence all areas of an image so that conflicts occur where lacking boundary information(low-frequency signals) especially in the black areas. The authors proposed the Minimal Impact ControlNet(MIControlNet) that can handle conflicts when different control signals are expected to manage different parts of the image. The introduced silent control signals which utilizes segmentation masks can rebalance the distribution of high-frequency information in the blank area. The authors reorganized the formula of ControlNet giving theoretical enhancement and ensuring more stable learning dynamics. The proposed multi-objective optimization strategies which utilizes Multiple Gradient Descent Algorithm (MGDA) improved the model performance.

**Strengths:**

The problem settings and overall paper structure are well organized.

The qualitative results for both single control signal and multi control signals utilizing segmentation masking are well aligned with handling the distribution of high-frequency information in the blank area. The blur effect shows in ControlNet even with single condition generation in silent area, however the MIControlNet shows more high-frequency informations in silent area.

The proof that gradient of the conservativity loss of ControlNet is equal to the gradient of the estimated conservativity loss and simplified loss for the ControlNet optimization are well explained.

The proposed model achieves the best FIDs in most cases within the multi-control signals.

**Weaknesses:**

In 2.2, the notations and symbols are confused and hard to read.

The proposed estimated conservativity loss function needs to construct second order derivatives, which is computationally expensive.

The authors proposed three main contributions with rebalancing the data distribution, feature injection and combination, and conservativity loss, however there are no ablation studies at each component in the paper. The only comparison is that both the original feature combination and a balanced version together between ControlNet and proposed model. For example, there are no comparison about Jacobian Asymmetry between the other baselines in Figure 4.(b). To claim the main contributions of each method, it would be beneficial to compare the performance improvements of each method against ControlNet.

The problem settings and the flow of the solutions are well-structured, however, the segmentation masking and the feature injection process shown in Figure 3 seems to be a little simple. Although the two main contributions(silent control signals, feature injection method) enhanced the model performance in the targeted problems, the method to apply the techniques are not quite novel.

Minor:

In Figure 6, the legend seems to have lack of information.

**Questions:**

In line 121, $f^e_c$ seems weird. Doesn't the value under $f$ mean layer number?

---

> ### Author Response · Authors · 2024-11-23
>
> We sincerely thank reviewer DjNC for acknowledging the relevance of our problem settings and the soundness of our work, particularly the theoretical aspects. Your suggestion to provide clearer ablation studies has greatly improved the clarity, comprehensibility, and overall completeness of this work.
> ## W1: The second stage is computationally expensive.
>
> Yes, the conservativity loss requires constructing a second-order computational graph, which is computationally expensive. However, we employ the following strategies to improve efficiency in our experiments:
> *  **FP16 Precision**: We use FP16 precision, which, although not natively supported for second-order computational graphs on `accelerate`, effectively reduces memory usage and computation time.
> * **Simplified Loss Function**: We develop a simplified loss function, $\mathcal{L}_{QC}^{simple}$, which **avoids constructing a second-order computational graph for the encoder** part of the U-Net.
> * **Selective Flash Attention**: Since Flash Attention does not currently support backpropagation through second-order computational graphs, we apply Flash Attention only to components that do not require gradient backpropagation for a second-order graph.
> * **Efficient Training Steps**: The training consists of only 2000 steps.
> As a result, the second-stage training requires approximately **7 hours**, which is entirely reasonable compared to the **48+ hours** needed for the first-stage training.
> ## W2: Need a clear version of ablation.
>
> We apologize for not providing a clear ablation analysis of our three contributions earlier. To address this, we first added the $Asym$ component for the baselines in Figure 4.(b) as part of the second experiment in **Common Concern 1**. Furthermore, we have included a comprehensive ablation study, as detailed in **Common Concern 2**.
> ## W3: Some other details.
>
> * We will refine the notations and symbols to enhance clarity and understanding. Details related to layers will be hidden in the main text, with comprehensive explanations provided in the appendix for those seeking further information.
> * We apologize for any inconvenience caused. In Figure 6, the shape represents the method, and the color represents the condition. We will revise and repaint the figure to make it more readable.
> *  In line 121, the value under $f$ indeed represents the layer number.

---

> > ### Comment · Reviewer_DjNC · 2024-11-24
> >
> > Thank you for your valuable response. The author’s answers have addressed my primary concerns. Therefore, I will maintain my score to 6.

---

> > > ### Author Response · Authors · 2024-11-26
> > >
> > > We sincerely express our heartfelt gratitude for your suggestion regarding the second-stage clarification and a clearer ablation. Your input has significantly enhanced the quality of our work. We remain open to further discussion.

---

### Official Review · Reviewer_ZUxW · 2024-11-03

**Soundness:** 4
**Presentation:** 3
**Contribution:** 3
**Rating:** 6
**Confidence:** 4

**Summary:**

- The paper introduces Minimal Impact ControlNet (MIControlNet), a novel framework designed to refine the integration of multiple control signals within diffusion models for image generation.

- The innovative approach of MIControlNet includes rebalancing data distribution, multi-objective feature combination, and reducing asymmetry in the Jacobian matrix to minimize conflicts between control signals.

- Experimental results demonstrate that MIControlNet outperforms baselines, achieving better compatibility and fidelity in generated images with multiple control signals.

**Strengths:**

- The method improves the compatibility of multiple control signals, which is crucial for applications requiring simultaneous control from different sources.

 - By addressing conflicts between control signals, the approach leads to higher quality image generation, particularly in areas with silent control signals.

**Weaknesses:**

- In Figure 1, why is it not the canny signal that suppresses the openpose signal? Is there some analysis?

- The authors should supplement corresponding **prompts**. Detailed prompts often lead to the generation of complex textures. Comparing the total variance and visual results under detailed/brief/null text conditions would enhance the persuasiveness of the proposed method in this paper.

A good response to the Weaknesses will improve my initial rating.

**Questions:**

See Weaknesses.

---

> ### Author Response · Authors · 2024-11-23
>
> We are deeply grateful to reviewer ZUxW for recognizing the significance of the issues we aim to address and acknowledging the strong performance of our methods. Your valuable assistance in **conducting experiments across detailed, brief, and null conditions has greatly enhanced the persuasiveness and robustness of our proposed approach.**
> ## W1: Why is it not the canny signal that suppresses the openpose signal?
>
> This is an important question, and we apologize for not explaining it more clearly. First, this phenomenon is supported by our experimental results. We believe the underlying cause lies in the feature injection and combination process. Specifically, in regions that should be influenced by the canny control signal, the openpose “silent” control signal may take precedence, effectively “speaking louder” than the Canny signal. Consequently, these regions generate content that aligns more closely with the OpenPose “silent” control signal.
>
> ## W2-1: Lack of the corresponding prompts.
>
> We apologize for not including detailed information about the prompts. For single-condition scenarios, the prompts are taken directly from the dataset, while for multi-condition scenarios, we concatenate the prompts for each condition. In the final version of our work, we will provide all the prompts used for the conditions.
>
> Specifically, regarding Figure 5, we have extended it into Figures 7 and 8 and included the related prompts for better clarity and completeness.
>
> ## W2-2: Add comparison with detailed/brief/null text conditions. on TV and visual results.
>
> This is an excellent suggestion for evaluating our work. Regarding the TV and FID comparison, we have provided detailed experimental results and analysis in the first additional experiments addressed in **Common Concern 1.**
>
> For the visual results, we present a more detailed comparison in Figures 7 and 8 on page 18 of the updated PDF. These figures illustrate examples under detailed, brief, and null text conditions, highlighting the corresponding visual outputs. From this comparison, we have observed the following key findings:
> * To implement the conservativity loss, we use SD 1.5 as the base model. However, **under null text conditions**, CFG is not applicable, leading to less visually appealing results. Despite this limitation, **our model clearly demonstrates the ability to generate textures effectively under the silent control signal**, whereas the original ControlNet fails to achieve this.
> * More detailed text conditions lead to better visual results overall.
> * **Our conservativity loss improves the model’s alignment with prompts**. For instance, in the Figure 7, our 2-stage model generates a snow-filled background, and in the Figure 8, the bread appears more complete.

---

> > ### Comment · Reviewer_ZUxW · 2024-11-26
> > **Thanks for the rebuttal**
> >
> > Thank you very much to the authors for their rebuttal and recognition of the review work. After reading the authors' rebuttal, the updated PDF, and the comments of other reviewers, I believe the authors have effectively addressed my concerns, and the effectiveness, credibility, and completeness of the methods presented in the paper have been further elaborated. Therefore, I am pleased to raise my score.

---

> > > ### Author Response · Authors · 2024-11-26
> > >
> > > We sincerely appreciate your recognition of our rebuttal! We are deeply grateful for your suggestion, and we believe that incorporating ablation studies over null, brief, and detailed prompts has significantly enhanced the quality of our manuscript. We remain open to further discussion and feedback.

---

### Official Review · Reviewer_ESQy · 2024-11-05

**Soundness:** 3
**Presentation:** 3
**Contribution:** 3
**Rating:** 6
**Confidence:** 3

**Summary:**

This paper presents a well-designed approach to multiple CtrlNet integration by addressing the challenge of control signal conflicts through a "minimal impact" strategy.
Specifically, the authors address several issues they found in multi-ControlNet integration.
1. To handle data bias in silent control regions, they use data augmentation and segmentation masks. This adds texture to low-signal areas.
2. To avoid conflicts between control signals, they use an MGDA-inspired feature injection method. This balances gradients so that no single control dominates.
3. They introduce a conservativity loss function to ensure stable training. This function keeps CtrlNet's influence to original U-Net minimal and improves consistency in multi-control outputs.

**Strengths:**

The solution proposed is clear and systematic. Experiments are comprehensive and convincing, covering various control signal combinations and providing robust quantitative and qualitative results. Limitation is discussed in the appendix.

**Weaknesses:**

Based on the methods presented, it appears that the proposed framework effectively supports scenarios involving more than two CtrlNets. Did the authors conduct experiments to evaluate performance in such cases? Since the experiments primarily focused on pairs of CtrlNets, I am curious about how well the approach scales to accommodate multiple CtrlNets.

Additionally, the authors mention the "Conservativity of Conditional Score Function", which seems to be a general principle that influences ControlNet performance, regardless of whether there is one or multiple CtrlNets in play. I wonder if the solutions proposed in this context could also enhance performance in standard single-CtrlNet applications. It might be beneficial for the authors to discuss this potential in the paper.

**Questions:**

Please refer to the *weakness* section.

---

> ### Author Response · Authors · 2024-11-23
>
> We extend our heartfelt thanks to reviewer ESQy for recognizing the clarity and systematic nature of our solution, as well as the robustness of our comprehensive experiments. Your valuable input on refining the experiments for single conditions has significantly enhanced the completeness of this manuscript. Additionally, your insightful suggestions have provided us with valuable direction for exploring future work.
> ## W1: Evaluate the performance in more than two conditions.
>
> This is an excellent suggestion for future work. However, **addressing more than two conditions** poses a significant challenge: unlike the two-condition case, **a straightforward analytical solution is not readily available.** Instead, iterative methods are typically employed to approximate or achieve the final combination. We are currently investigating the use of the Frank-Wolfe Algorithm[1] to tackle this issue. Upon completing the experiments, we plan to include the corresponding results and analysis in the final version of our work.
> ## W2: Conservativity loss on single condition.
>
> In response to **Common Concern 1**, we present detailed experimental results and analysis from the first experiment, comparing MIControlNet (1-stage) with MIControlNet (2-stage) to perform an ablation study on the conservativity loss.
>
> [1] Sener, Ozan, and Vladlen Koltun. "Multi-task learning as multi-objective optimization." _Advances in neural information processing systems_ 31 (2018).

---

### Author Response · Authors · 2024-11-23
**Rebuttal to all reviewers**

## Common Concerns 1: More experiments.

**1. Calculate FID and Total Variance For ControlNet, MIControlNet(1-stage) and MIControlNet(2-stage).**

Below is the table presenting the FID and Total Variance (TV, in units of $1\times 10^4$) for Canny and OpenPose conditions under three scenarios: no prompts, brief prompts, and detailed prompts. The following summary tables provide an overview of the results.

| **(FID, TV)**                 | **ControlNet** | **MIControlNet(1-stage)** | **MIControlNet(2-stage)** |
| ----------------------------- | ---------------- | ------------------------- | ------------------------- |
| **Canny No Prompts**          | 109.6, 2.62      | 114.4, 3.54               | **123.9, 3.79**           |
| **Canny Brief Prompts**       | 89.34, 2.48      | 89.77, 3.24               | 90.18, 3.15               |
| **Canny Detailed Prompts**    | 88.55, 2.47      | 90.21, 3.28               | 89.37, 3.36               |
| **OpenPose No Prompts**       | 132.5, 3.34      | 131.9, 3.39               | **133.0, 3.67**          |
| **OpenPose Brief Prompts**    | 97.08, 2.52      | 98.32, 2.71               | 98.14, 2.74               |
| **OpenPose Detailed Prompts** | 99.09, 2.70      | 95.34, 2.92               | 94.16, 2.91               |
We have the following findings:

* MIControlNet, with or without the conservativity loss, **demonstrates similar FID performance**. However, **with conservativity loss, MIControlNet exhibits improved pattern generation ability** under silent control signals, **as highlighted in the table above.**
* MIControlNet achieves **comparable FID performance** to the baseline but **demonstrates significantly stronger performance in terms of total variance.**
* When comparing **no prompts**, **brief prompts**, and **detailed prompts**, **providing more detailed prompts generally leads to better FID performance and smaller total variance.**
* Interestingly, **for detailed prompts, the total variance tends to slightly increase**. We hypothesize that this is due to the **more detailed prompts offering finer control under silent control signals**, thereby generating more diverse patterns.


**2. The $Asym$ for Regular ControlNet**

We calculate the $Asym$ for Regular ControlNet and compare it with our MIControlNet(1-stage) and MIControlNet(2-stage), the results are shown in the following table:

| Condition              | Canny  | Hed    | OpenPose |
| ---------------------- | ------ | ------ | -------- |
| ControlNet             | 56.75  | 22.41  | 6.454    |
| MIControlNet (1-stage) | 29.87  | 38.28  | 3.980    |
| MIControlNet (2-stage) | 0.1174 | 0.1894 | 0.0274   |
We have the following findings:

* MIControlNet (1-stage) performs slightly better than ControlNet in terms of $Asym$.
* MIControlNet (2-stage) significantly outperforms both ControlNet and MIControlNet (1-stage) on $Asym$.
* For each condition, MIControlNet(2-stage) demonstrates consistent improvements on a logarithmic scale.

## Common Concerns 2: A more clear ablation.

The FID for a throughly ablation:

| Method                                | Openpose-Canny        | Canny-Hed             | Hed-Depth             |
| ------------------------------------- | --------------------- | --------------------- | --------------------- |
| Vanilla ControlNet                    | 80.37 / 111.30        | 123.59 / 86.43        | 91.98 / 86.25         |
| + Data Augmentation                   | 92.98 / 84.02         | 77.02 / 75.46         | 74.28 / 81.16         |
| + Our Feature Injection & Combination | 76.13 / 77.22         | 74.19 / 70.26         | 71.16 / 71.93         |
| + Conservativity Loss                 | **75.77** / **72.25** | **71.34** / **69.35** | **69.68** / **71.18** |
We observe the following:

* Our silent control signal-targeted data augmentation, feature injection & combination, and conservativity loss all lead to improvements in FID scores.
* The conservativity loss, particularly for Canny combined with other conditions, achieves a consistent improvement of approximately 3 points in FID.
* The improvements achieved through the conservativity loss are consistent, and we have further strengthened its theoretical foundation, particularly in the context of modular neural networks designed to **optimize GPU memory usage and computational efficiency**.

---

### Author Response · Authors · 2024-11-23
**Rebuttal to all reviewers**

We sincerely thank all the reviewers for their invaluable feedback and contributions. We also extend our apologies for the delayed response.

*  We extend our heartfelt thanks to reviewer ESQy for recognizing the clarity and systematic nature of our solution, as well as the robustness of our comprehensive experiments. Their valuable input on refining the experiments for single conditions has significantly enhanced the completeness of this manuscript. Additionally, **their insightful suggestions have provided us with valuable direction for exploring future work.**
* We are deeply grateful to reviewer ZUxW for recognizing the significance of the issues we aim to address and acknowledging the performance of our methods. **Their valuable assistance in conducting experiments across detailed, brief, and null conditions has greatly enhanced the persuasiveness and robustness of our proposed approach.**
* We sincerely thank reviewer DjNC for acknowledging the relevance of our problem settings and the soundness of our work, particularly the theoretical aspects. **Their suggestion to provide clearer ablation studies has greatly improved the clarity, comprehensibility, and overall completeness of this work.**
* We extend our heartfelt gratitude to reviewer TnUC for recognizing the depth of our problem analysis and the theoretical contributions of our work. **Their suggestions helped us include additional details in the preliminaries and provide a clearer explanation of the conservativity loss**, enabling our work to reach and resonate with a broader audience.

Thank you all very much again for your thoughtful and constructive feedback.

---

### Meta-Review · Area_Chair_iDPN · 2024-12-16

**Metareview:**

This paper introduces Minimal Impact ControlNet (MIControlNet), a novel method to address the challenges of integrating multiple control signals in diffusion models for image generation. The authors identify conflicts arising from "silent control signals" in ControlNet, which can suppress texture generation in certain image areas. To mitigate these conflicts, they propose three strategies: rebalancing the training data, combining and injecting feature signals in a balanced manner, and addressing asymmetry in the score function's Jacobian matrix.

Strengths:
- Clear and systematic solution to the problem of multi-ControlNet integration.
- Comprehensive experiments covering various control signal combinations with robust quantitative and qualitative results.
- Effective analysis of the challenges in using multiple control signals, particularly the issue of "silent control signals".
- Theoretical contribution and application of conservativity loss on a modular network.

Weaknesses:
- Initial lack of clarity in some phrases and missing citations.
- Need for improved organization and more detailed ablation studies.
- Computational cost of the proposed method, especially the second-order computational graph.

Despite the initial weaknesses, the authors have effectively addressed the reviewers' concerns during the discussion phase. They have improved the clarity of the manuscript, added missing citations, and conducted more comprehensive ablation studies. The proposed method shows clear improvements in managing ControlNet conflicts and allows for more harmonious image generation using multiple control signals. The theoretical contributions and the practical implications of this work make it a valuable addition to the field of generative models.

**Additional Comments On Reviewer Discussion:**

During the rebuttal period, reviewers raised several concerns, including the need for more experiments, clearer ablation studies, and improved clarity in writing. The authors responded positively to these concerns and made significant changes to the manuscript. They added experiments for single conditions, provided detailed ablation studies, and clarified confusing phrases and notations. These revisions have strengthened the paper and addressed the reviewers' concerns effectively.

---

### Decision · Program_Chairs · 2025-01-22

Accept (Poster)